# Resolving the origins of secretory products and anthelmintic responses in a human parasitic nematode at single-cell resolution

Clair R Henthorn[1], Paul M Airs[1], Emma K Neumann[2], Mostafa Zamanian[1]*

[1]Department of Pathobiological Sciences, University of Wisconsin-Madison, Madison, United States; [2]Pathology and Laboratory Medicine, School of Medicine and Public Health, University of Wisconsin-Madison, Madison, United States

**Abstract** Nematode excretory-secretory (ES) products are essential for the establishment and maintenance of infections in mammals and are valued as therapeutic and diagnostic targets. While parasite effector proteins contribute to host immune evasion and anthelmintics have been shown to modulate secretory behaviors, little is known about the cellular origins of ES products or the tissue distributions of drug targets. We leveraged single-cell approaches in the human parasite *Brugia malayi* to generate an annotated cell expression atlas of microfilariae. We show that prominent antigens are transcriptionally derived from both secretory and non-secretory cell and tissue types, and anthelmintic targets display distinct expression patterns across neuronal, muscular, and other cell types. While the major classes of anthelmintics do not affect the viability of isolated cells at pharmacological concentrations, we observe cell-specific transcriptional shifts in response to ivermectin. Finally, we introduce a microfilariae cell culture model to enable future functional studies of parasitic nematode cells. We expect these methods to be readily adaptable to other parasitic nematode species and stages.

**\*For correspondence:**
mzamanian@wisc.edu

**Competing interest:** The authors declare that no competing interests exist.

## Editor's evaluation

In this important study, the authors generate and analyse single-cell datasets for the human parasitic nematode Brugia malayi. The new resource has the potential to uncover new details of the biology of secretory systems in this filarial nematode. The analyses are of high quality but some concerns remain about the quality of parts of the data due to difficulties inherent in obtaining and preparing samples from this parasite. The new resource will be of broad interest to parasitologists and nematode biologists and has the potential to accelerate research in the search of new anthelmintics and vaccines.

## Introduction

Lymphatic filariasis (LF) is a chronic and debilitating neglected tropical disease caused by the filarial nematodes *Wuchereria bancrofti, Brugia malayi,* and *Brugia timori*. Infective-stage larvae are transmitted by mosquitoes to the human definitive host, where they develop and molt to adult stages that persist in the lymphatics and produce blood-circulating pre-larval stage microfilariae (mf) (*Roberts et al., 2009*). LF afflicts an estimated 51 million individuals in tropical and subtropical climates around the world and endangers nearly a billion individuals worldwide (*Local Burden of Disease 2019 Neglected Tropical Diseases Collaborators, 2020*; *Local Burden of Disease 2019*

*Neglected Tropical Diseases Collaborators, 2020*; *World Health Organization, 2018*). Chronic cases of LF manifest in elephantiasis, an extreme buildup of lymph resulting in stigmatizing disfiguration and additional socioeconomic challenges (*Ton et al., 2015*; *Weiss, 2008*). Mass drug administration (MDA) in endemic settings with combinations of ivermectin (IVM), diethylcarbamazine (DEC), and albendazole (ABZ) is used to disrupt parasite transmission, but this approach is only effective with years of repeated treatment. Additionally, contraindications for antifilarial drugs exist in regions co-endemic for other filarial parasites (*Chippaux et al., 1996*). With no cure and the growing threat of resistance to anthelmintics (*Campbell, 1982*; *Geary et al., 2011*; *Geary et al., 2010*; *Ismail et al., 1999*; *Osei-Atweneboana et al., 2011*; *Wolstenholme et al., 2015*), there is a clear need to improve our understanding of basic biological processes that underpin the host–parasite interaction.

The excretory-secretory (ES) products of parasitic nematodes are known to be essential for parasite survival within the host (*Harnett, 2014*; *Hotterbeekx et al., 2021*; *Lightowlers and Rickard, 1988*). Despite the general understanding that the ES system is a conduit for the release of immunomodulatory proteins and extracellular vesicles that promote parasite survival, the biology of the ES system has yet to be investigated in medically important parasitic nematodes. Profiling of secretions (*Bennuru et al., 2011*; *Harischandra et al., 2018*; *Hewitson et al., 2008*; *Kaushal et al., 1982*; *Moreno and Geary, 2008*; *Zamanian et al., 2015*) across the filarial nematode life cycle has helped identify antigens that have been pursued therapeutically and diagnostically (*Harnett, 2014*; *Kalyanasundaram et al., 2020*; *Maizels et al., 2001*; *Morris et al., 2013*), but the origins of these proteins and the tissue systems that underlie their release at the host–parasite interface are unknown. Recent studies have also implicated secretory processes as a target of existing anthelmintics (*Moreno et al., 2021*; *Moreno et al., 2010*). IVM causes rapid clearance of mf from host blood at concentrations that do not cause overt fitness effects on cultured parasites. This disconnect is reconciled by a model whereby IVM unmasks mf from the host immune system by inhibiting protein and vesicle secretion (*Rao et al., 1987*; *Vatta et al., 2014*; *Zahner et al., 1997*). While the ES apparatus is thought to be the primary source of immunogenic proteins in the mf stage, the cells that comprise this structure and control secretion have not been characterized.

Clear morphological descriptions of the filarial ES system are currently restricted to the mf stage, but microscopy studies confirm that this structure is present in L3 and adult parasites (*Airs et al., 2022*; *Landmann et al., 2010*; *Mutafchiev et al., 2014*; *Tongu, 1974*). Filarial parasites possess tubular ES systems featuring a single large excretory cell connected through a cytoplasmic bridge to an excretory vesicle and pore (*Tongu, 1974*). In the free-living model nematode *Caenorhabditis elegans*, the ES system is a well-studied and critical osmoregulatory and excretory organ made up of a pore cell, duct cell, excretory/canal cell, and canal-associated neurons (CAN) (*Sundaram and Buechner, 2016*). However, significant diversification of ES anatomy and function between *C. elegans* and filariae and more generally across nematode clades (*A. F. Bird, 1991*) demands more species-specific approaches to the study of ES systems.

Single-cell transcriptomics has facilitated resolution of cellular processes and functions that are masked in tissue level or bulk transcriptomic approaches. Single-cell suspensions generated from tissue dispersions of free-living nematodes (*Zhang et al., 2011*) have enabled high-dimensional transcriptomic analysis of cell types across developmental stages (*Ben-David et al., 2020*; *Cao et al., 2017*; *Packer et al., 2019*; *Taylor et al., 2021*; *Zhang et al., 2011*) and provide a pathway to the characterization of essential cells and tissues in related parasites, including those that control secretory behaviors. However, viable cell suspensions have yet to be leveraged to transcriptomically profile human, animal, or plant parasitic nematodes at single-cell resolution. This goal is complicated in many species by life cycle patterns that can limit tissue availability and anatomical variation that can affect access to rare cell types.

Here, we applied single-cell approaches to generate and annotate a cell atlas of gene expression in *B. malayi* microfilariae and to map secretory-associated cell types and the distributions of prominent antigens and anthelmintic targets. We evaluated the effects of anthelmintics on parasite cells and the amenability of mf cell suspensions to longer-term culture. These data and methods allowed for novel inferences about the origins of immunogenic molecules, the mechanism of action of existing anthelmintics, and provide an avenue for future functional studies of *B. malayi* cell populations. We expect that many of these tools can be extended to other medically important parasitic nematodes.

## Results

### Generation of viable single-cell suspensions from *B. malayi* microfilariae

We focused our efforts on generating single-cell suspensions from the blood-circulating mf stage of the human parasitic nematode *B. malayi*. This pre-larval life stage is small (~177–230 µm in length, ~5–7 µm in diameter) but can be isolated from peritoneal fluid of infected jirds in large quantities (millions), potentially facilitating insight into early parasite development and the capture of rare cell types. Using larval tissue dispersion protocols from *C. elegans* as a scaffold, we developed a single-cell dissociation protocol in *B. malayi* mf that accounts for the unique biology of this parasite stage (*Figure 1A*). Microfilariae suspensions recovered from the jird peritoneal cavity are developmentally asynchronous and include an abundance of host cells, tissue debris, embryos, and nonviable mf, all of which contribute to transcriptomic contamination. To reduce contamination and enrich for viable mf, peritoneal lavages were filtered using Sephadex PD-10 desalting columns (*Galal et al., 1989*; *Rathaur et al., 1987*). This method reliably recovered high yields (60–70%) of highly motile mf that were then used as input into tissue dispersion reactions (*Figure 1B*).

*B. malayi* mf are encapsulated within an impermeable eggshell of carbohydrates including chitin (*Fuhrman and Piessens, 1985*). We treated filtered mf with chitinase to penetrate this sheath and allow downstream dispersion reagents to access the underlying cuticle and worm body. Chitinase treatment did not cause complete exsheathment, but produced a visible shrinking effect in the tail and head spaces indicating sheath breach (*Figure 1C*). SDS-DTT treatment was optimized to weaken protective structures (sheath and cuticle) of chitinase-treated worms while preserving cell health. Incubation in diluted SDS-DTT effectively compromised cuticular structure and inhibited motility without causing parasite death (*Figure 1—figure supplement 1*). Recoverable inhibition of motility, as revealed by subsequent wash steps, served as visual confirmation of the efficient weakening of the cuticle and preservation of the underlying body.

The generation of single-cell suspensions was completed using pronase enzymatic digestion and mechanical disruption by continuous pipetting. Progression of the pronase digestion was monitored throughout the treatment (*Figure 1D*). mf cuticles were visibly compromised after approximately 20 min of digestion, where the midsection of the filarial body plan exhibits protrusions that break and release cells. Progression of the reaction yields a mixed suspension of worm segments, single cells, and intact mf, and an abundance of single cells can be seen by 30 min. Continuous mechanical disruption was essential to encourage the breakage of worms and release of single cells. The majority of undigested worms were removed by centrifugation and filtration, resulting in a highly viable single-cell mf suspension for downstream applications.

### High-content imaging of cell suspensions confirms capture of rare parasite cell types

The integrity of cells retrieved from *Brugia* mf dispersions was assessed by imaging flow cytometry. A total of 20,000 objects were assessed and 15,683 were identified as debris, undigested worm segments, or nonviable cells as indicated by negative Calcein-AM staining, a cell-permeable viability dye that remains within healthy cells after acetoxymethyl (AM) cleavage by non-specific esterase activity. The remaining 4317 objects were analyzed using post-acquisition image features to identify 2893 objects as live, single cells that displayed round cellular and nuclear morphology (*Figure 1E*). While the majority of recovered cells have a diameter of 5.23 ± 0.77 µm, each mf is known to possess a large and morphologically distinct secretory cell that is critical to parasite survival and immune evasion in the host (*Tongu, 1974*; *Hewitson et al., 2008*). The *Brugia* secretory cell is hypothesized to be responsible for the release of prominent antigens, extracellular vesicles, and nucleic acids that operate at the host-parasite interface. Classical descriptions of the mf ES system highlight the secretory cell as a large cell with bipolar canal extensions (*Moreno et al., 2010*; *Nelson and Riddle, 1984*; *Sundaram and Buechner, 2016*). We mined cells in our dataset that match this description and identified a total of 11 cells with bipolar appendages that are notably the largest identifiable cells (8.46 ± 0.77 µm in diameter) in this life stage (*Figure 1E*). Each mf possesses a single secretory cell among hundreds of total cells, and this recovery rate (0.4%) falls within the expected range, providing us confidence that our dispersion protocol captures both common and rare cell types.

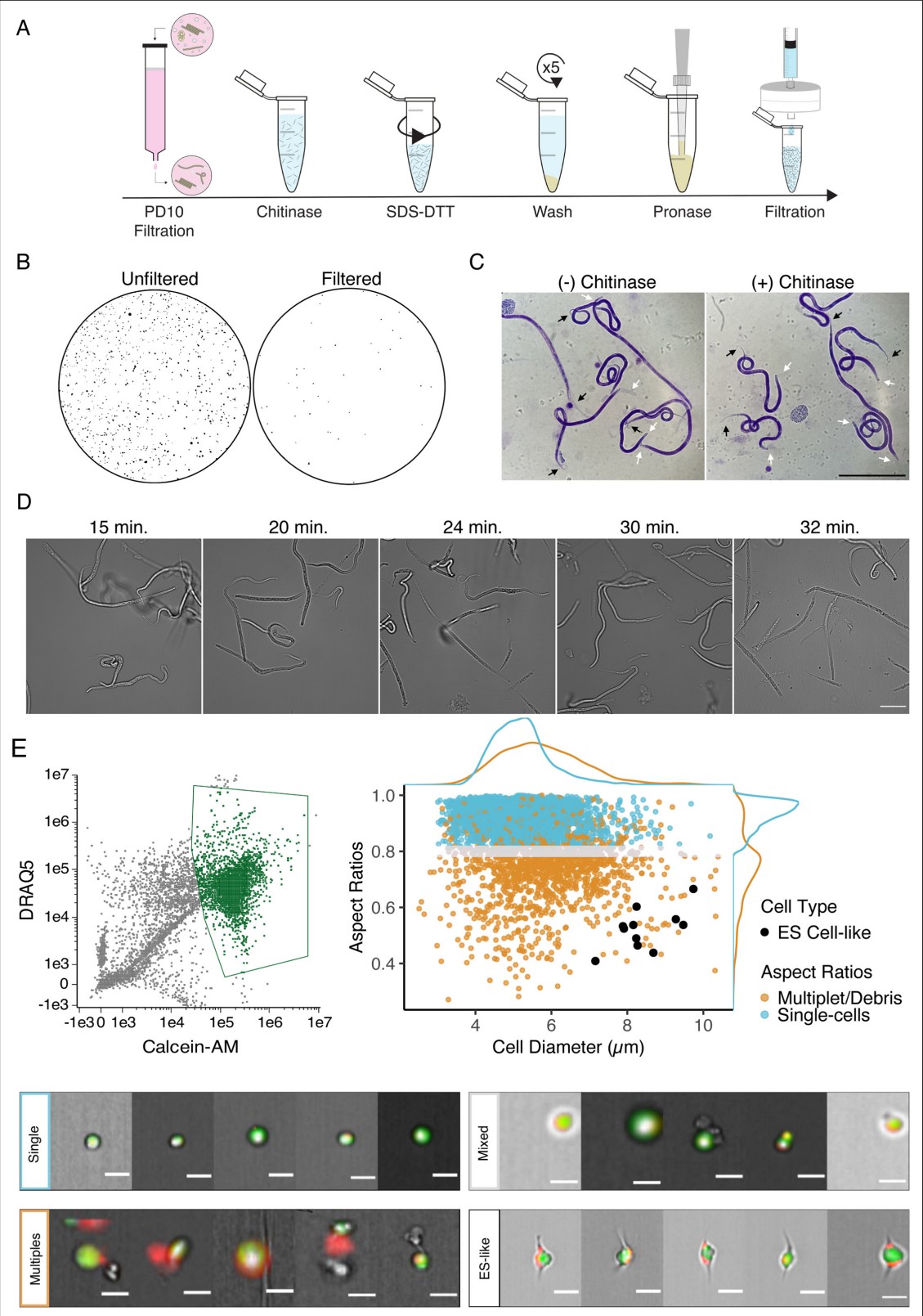

**Figure 1.** Optimization of single-cell dissociation in *B. malayi* microfilariae. (**A**) Schematic of dissociation protocol using 1 million *B. malayi* microfilariae as input. (**B**) Images of 96-well plate wells containing pre- and post- PD-10 filtration of microfilariae-containing peritoneal lavage. Unfiltered solutions (left) contain a plethora of host cell and tissue contamination and parasite embryos. Filtered solutions (right) minimize host contamination carryover prior to cell dissociation and downstream applications. (**C**) Representative Giemsa stained microfilariae pre-chitinase (left) and post-chitinase (right) treatment.

*Figure 1 continued on next page*

*Figure 1 continued*

Chitinase disrupts sheath integrity visible in the head (white arrows) and tail spaces (black arrows). Scale bar = 100 μm. (**D**) Cell dissociation timelapse during pronase digestion and mechanical disruption. Scale bar = 50 μm. (**E**) High-content imaging flow cytometry of single-cell suspensions. Left: gating scheme to identify objects based on nucleation (DRAQ5+ stain) and high viability (Calcein-AM+ stain). Objects colored in green indicate highly viable, nucleated objects carried forward in the downstream analysis. Right: single-cell objects were segregated by IDEAS software and objects with aspect ratios ≥ 0.8 for brightfield and DRAQ5 channels were counted as single-cells (blue). A mixed population (gray) included objects belonging to doublets/multiples/debris (orange). All captured images were scanned by eye for cells with excretory-secretory (ES)-like morphology (black) and included as single cells. Bottom: representative images of cells belonging to the identified single-cell (blue), mixed (gray), ES-like (black), and multiplet/debris (orange) populations. Scale bars = 10 μm.

The online version of this article includes the following source data and figure supplement(s) for figure 1:

**Source data 1.** Comparison of cell counting methods by hemocytometer, flow cytometry, and automated cell counting instrument.

**Figure supplement 1.** Optimization of SDS-DTT incubation by time and concentration.

## Single-cell transcriptomic atlas identifies conserved and unique pre-larval cell populations

To define the transcriptional profiles of *B. malayi* microfilariae cell types, we generated a single-cell RNA-seq (scRNA-seq) library from cell suspensions. Dimensional reduction of 46,621 filtered cells via Uniform Manifold Approximation and Projection (UMAP) (*McInnes et al., 2018*) identified 27 transcriptionally distinct clusters. Across all cells in the dataset, a median of 230 genes and 267 reads per cell were achieved (*Figure 2—figure supplement 1*). Transcript abundance of single cells strongly correlated with microfilariae bulk gene expression (*Reaves et al., 2018*; *Figure 2A*, inset, $r = 0.85$, $R^2 = 0.72$, $p=2.2 \times 10^{-16}$), providing additional evidence that our single-cell pipeline largely recapitulates the cell-type distribution and transcriptional profile of the intact parasite. An analysis of *B. malayi* transcription factors shows both specific and broad-based transcription factor activity that aligns with known *C. elegans* patterns (*Figure 2—figure supplement 2*). For example, *lin-14* exhibits broad expression in the *B. malayi* mf scRNA-seq data and is congruent with broad expression of the transcription factor in *C. elegans* (*Taylor et al., 2021*). In contrast, *C. elegans* has multiple cell-type-restricted transcription factors including ciliated sensory neuron transcription factor *Cel-daf-19* and body wall muscle-specific transcription factors *Cel-hlh-1* and *Cel-unc-120* (*Taylor et al., 2021*), both of which show restricted cluster expression in the *B. malayi* mf scRNA-seq atlas. These analyses provide confidence in the biological signal acquired through scRNA-seq. Markers for well-characterized *C. elegans* cell types (*Figure 2—figure supplement 3*) with one-to-one *B. malayi* orthologs were used to annotate 16 UMAP clusters (*Figure 2A*), and pseudobulk analysis confirms the robustness of these markers for differentiating cell types (*Figure 2—figure supplement 4*). In total 18,317 cells (39.3% of global count) were assigned a cell-type annotation. Among these, 6,223 cells were identified as muscle derived (13.3% of global count) and 5,527 cells expressed pan-neuronal markers (11.9% of global count). Additional identifiable clusters include those representing canal-associated cells, coelomocytes, mesodermal tissues, and a pre-alimentary canal-related cell type.

Cell types belonging to the mesoderm lineage include 10,284 cells (22% of global count). Body wall muscle cells represent the majority of the mesodermal cells identified (clusters 2 and 19) and are distinctly categorized based on expression of *Bma-hlh-1* (*Krause et al., 1994*) and markers shown to be enriched in *B. malayi* body wall muscle such as actin and multiple myosins (*Bma-myo-3*, *Bma-unc-54.1*, *Bma-unc-54.2*, *Bma-act-1*) (*Morris et al., 2015*). An additional two clusters (clusters 9 and 17) strongly express markers associated with mesodermal lineage *C. elegans* cell types (enteric and vulval), including *Cel-hlh-8*, *Cel-hlh-2*, and *Cel-mls-1* orthologs (*Packer et al., 2019*; *Philogene et al., 2012*). Differentially expressed genes between the two clusters are proteins of unknown function and could not be further parsed. Also belonging to the mesodermal cell lineage in *C. elegans* are coelomocytes, phagocytic scavengers located within the pseudocoel. Using coelomocyte-specific *C. elegans* orthologs (*Bma-cup-4*, *Bm6921*, *Bma-unc-122*, *Bma-let-381*), cluster 6 was annotated as coelomocytes. These data show that mesodermal cells in the post-embryonic and pre-larval mf state include well-differentiated body wall musculature and coelomocytes and underdeveloped enteric and vulval muscle structures.

Pan-neuronal *C. elegans* markers (*sbt-1*, *ric-4*, *ida-1*, *egl-3*, *egl-21*) were used to identify nine clusters of putative neurons expressing all five markers (303 genes and 358 reads per cell) (*Figure 2B*).

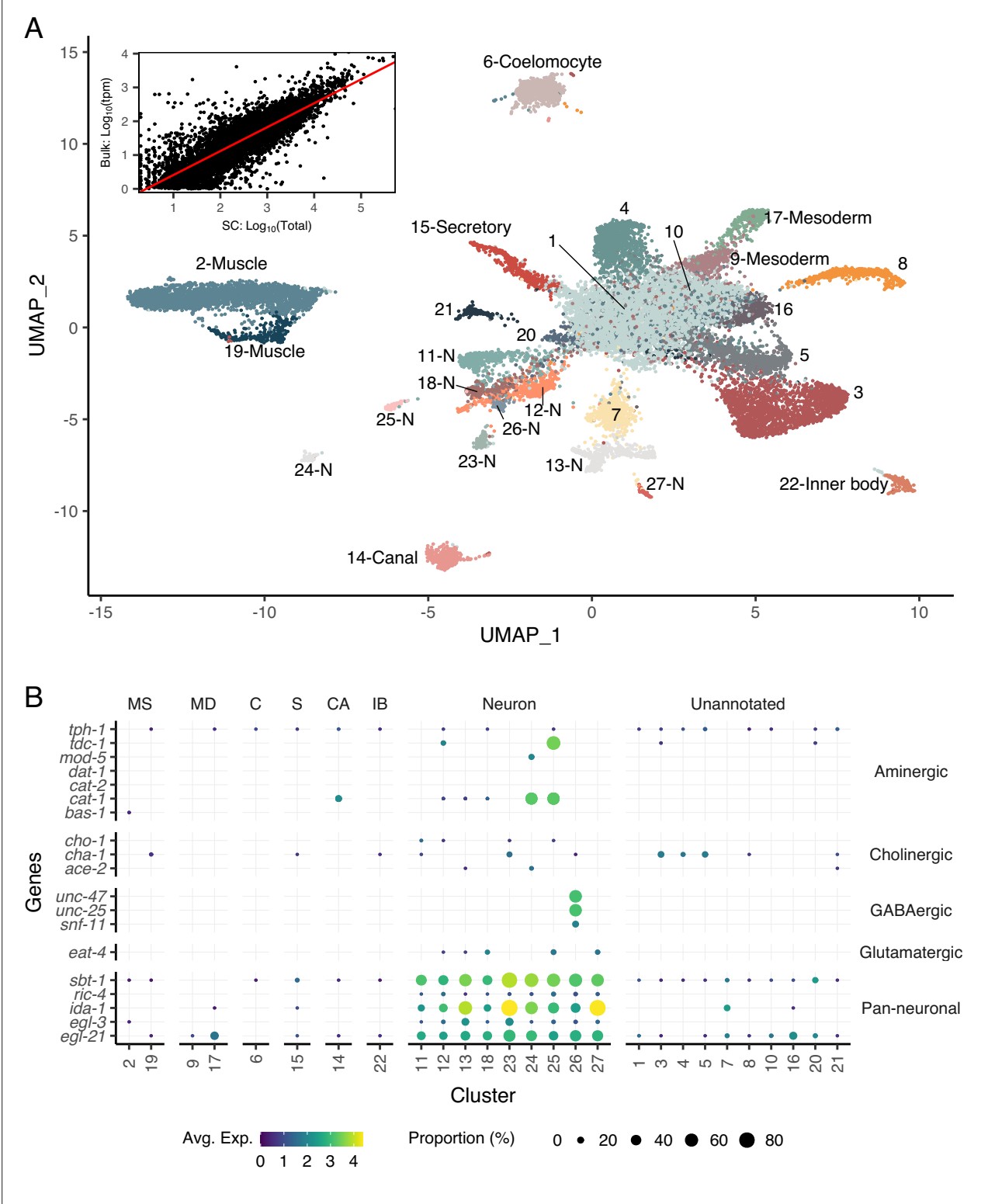

**Figure 2.** A single-cell transcriptomic atlas of *B. malayi* microfilariae cell types. (**A**) Global Uniform Manifold Approximation and Projection (UMAP) transcriptome clustering of 46,621 cells with cell-type annotations. Inset: comparison of bulk transcript per million counts (tpm) (bulk: $Log_{10}$(tpm)) and single-cell RNA-seq total read counts (SC: $Log_{10}$(Total)) indicates the transcriptomic profile of the single-cell atlas largely recapitulates the gene expression profile of bulk tissue RNA-sequencing. Correlation between the bulk and single-cell RNA-seq datasets was assessed using Pearson correlation coefficient. Linear model regression line is indicated in red. (**B**) Gene expression distribution (color) and proportion of cells per cluster (size)

*Figure 2 continued on next page*

*Figure 2 continued*

expressing neurotransmitter-specific genes. Clusters 24 and 25 are aminergic and cluster 26 is GABAergic. Glutamatergic and cholinergic neurons are unresolved. MS, muscle; MD, msoderm; C, coelomocyte; S, secretory; CA, canal-associated; IB, inner body.

The online version of this article includes the following figure supplement(s) for figure 2:

**Figure supplement 1.** Histograms of genes (left) and UMIs (right) per cell across all clustered cells.

**Figure supplement 2.** Distribution of transcription factors by cell type.

**Figure supplement 3.** Uniform Manifold Approximation and Projection (UMAP) visualization of marker genes used to annotate cell-type clusters.

**Figure supplement 4.** Pseudobulk analysis of scRNA-seq transcriptomic data.

**Figure supplement 5.** Distribution of neuronal gene transcripts across all clusters.

**Figure supplement 6.** Integration of single-cell datasets using Scanorama algorithm.

We were able to further resolve a cluster of motor neurons (cluster 11) and interneurons (cluster 23) characterized by one-to-one orthologs of the *C. elegans* motor neuron marker (*Cel-unc-4*) and DVA tail interneuron marker (*Cel-nlp-12*), respectively (*Figure 2—figure supplement 4*). Clusters 13 and 27 express *Bma-daf-19*, an ortholog of the RFX-family transcription factor *Cel-daf-19* shown to be directly responsible for cilia development in ciliated sensory neurons as well as genes involved in ciliogenesis (*Figure 2—figure supplement 3*; *Blacque et al., 2005*; *Chen et al., 2006*; *Narasimhan et al., 2015*; *Swoboda et al., 2000*). These two clusters have been annotated as ciliated sensory neurons. Known genes involved in neurotransmitter synthesis and transport were used to map neuropeptidergic (cluster 26) and aminergic neurons (clusters 24 and 25) (*Figure 2B*). Acetylcholine is the primary excitatory nematode neurotransmitter, and we used cholinergic pathway genes (*Bma-cha-1*, *Bma-cho-1*, and *Bma-ace-2*) to locate cholinergic neurons. The motor and interneuron neuronal clusters (11 and 23) express *Bma-cha-1* and *Bma-cho-1* in 10–15% of all captured cells. Clusters 3–5 exhibit more abundant expression of *Bma-cha-1* (*Figure 2—figure supplement 5*) but no other neuronal markers, cholinergic or pan-neuronal, and may represent a cholinergic neuronal developmental state.

The phylogenetic distance between free-living clade V *C. elegans* and filarial clade III nematodes is associated with known differences in anatomy (*Chitwood and Chitwood, 1950*) and localization of some orthologous proteins (*Moreno et al., 2010*), which limits efforts to comprehensively map all cell types across species (*Figure 2—figure supplement 6*). Cell types and structures unique to filarial nematodes demand different annotation approaches. A cluster of 503 cells (cluster 22) uniquely displays high expression of chitinases (*Bma-cht-1.1*, *Bma-cht-1.2*, *Bma-cht-1.3*) and the immunomodulatory vaccine candidates *Bma-val-1* and *Bm97*. Chitinase has been shown to be stored in the inner body of microfilariae that is positioned between the excretory and G1 cells and is hypothesized to act as the precursor for the intestinal tract of the developing larvae in the mosquito (*McLaren, 1972*; *Wu et al., 2008*). The combination of chitinase and molting proteins within this segregated cluster supports the annotation of cluster 22 as inner body-associated cells.

Among *C. elegans* excretory-secretory cell types, only canal-associated (CAN) cell-specific markers could be associated with a *B. malayi* single-cell cluster. *B. malayi* orthologs (*Bma-pks-1*, *Bma-acbp-6*, *Bma-ceh-10*) of CAN markers identify cluster 14 as ES CAN cells (1006 total cells). Given the unique cell composition and structural adaptations of the ES system in filarial nematodes, identification of additional secretory cell types demands other approaches.

## Identification of secretory-related cell types using FACS and RNA-seq approaches

We exploited the large size of the *B. malayi* secretory cell as a means to enrich for and profile this rare cell type within single-cell suspensions. The largest viable cells within a *B. malayi* microfilariae suspension were FAC-sorted and pooled for downstream RNA-seq (*Figure 3A* and *Figure 3—figure supplement 1*). The transcriptional profiles of these outlier cells were compared to smaller round cells that typify the mf cell population, and unique identifiers of the largest cells were mapped to the single-cell atlas UMAP. Cluster 15 shows a clear overall enrichment for the 78 genes more highly expressed in this sorted cell population (*Figure 3B* and *Figure 3—source data 1*), and individual cells within this putative secretory cell cluster represent the largest group coexpressing different subsets of these markers (*Figure 3C* and *Figure 3—source data 2*).

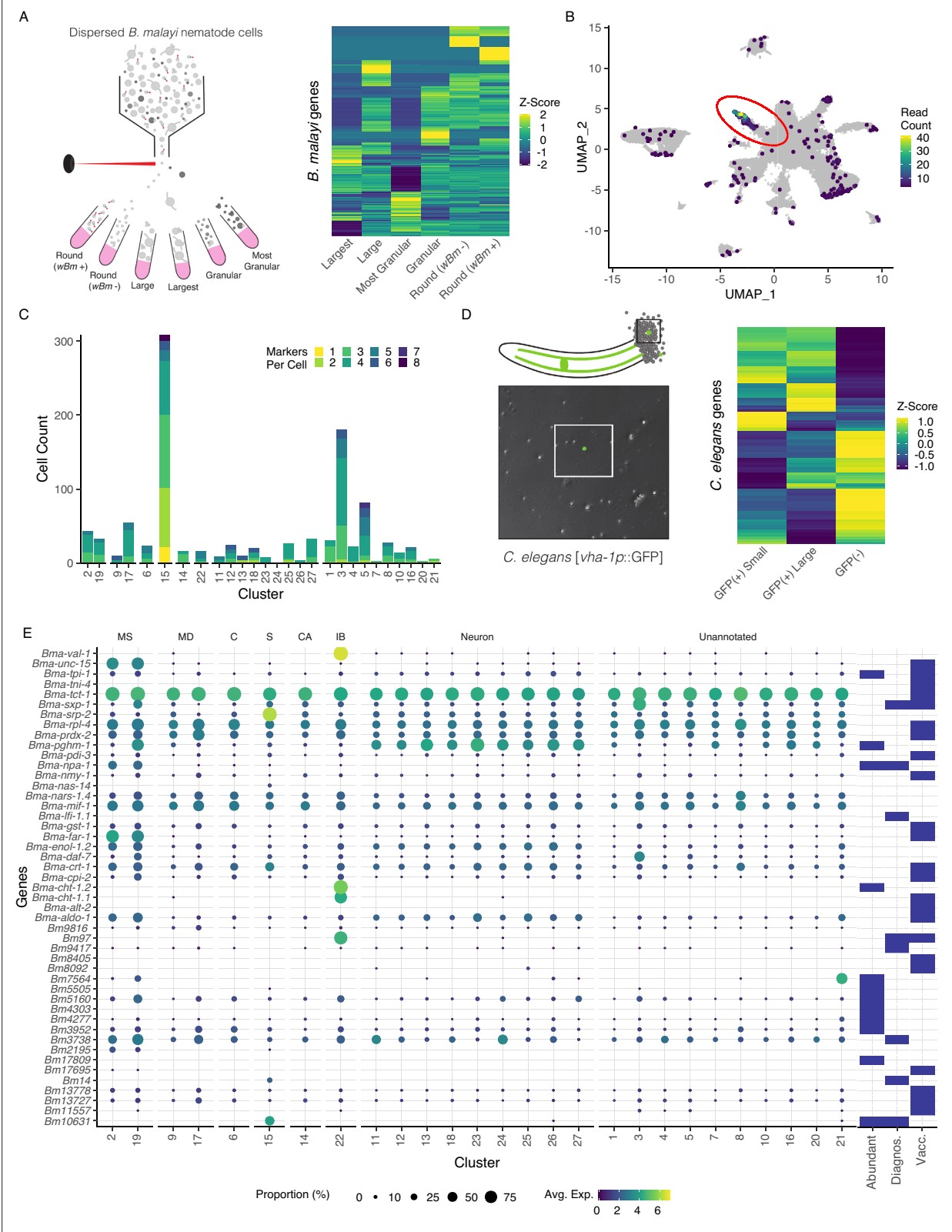

**Figure 3.** Annotation of the *Brugia* secretory cell and localization of secretory-related antigens indicates broad distribution of antigen transcriptic origins. (**A**) Schematic of fluorescent-activated cell sorting (FACS) enrichment approach for isolating cell populations in *B. malayi* microfilariae (mf) single-cell dispersions. Viable cells were sorted by size, granularity, and the presence or absence of anti-*Wolbachia* fluorescent antibodies and collected directly into TRIzol LS for RNA isolation and sequencing. (**B**) Differentially expressed genes (DEGs) in the 'Largest' cells sample, the population

*Figure 3 continued*

hypothesized to contain the secretory cell, were projected on the mf single-cell atlas and indicated expression enrichment in cluster 15. Read count represents summed reads across mapped genes in the single-cell atlas. (**C**) Cell and marker coexpression quantification in cells belonging to cluster 15 using the DEGs identified in (**A**). Cluster 15 includes the majority of cells expressing DEGs and contains the most cells coexpressing two or more markers. (**D**) Single-cell suspension of *C. elegans* strain BK36 with the excretory canal cytoplasm labeled with GFP. Heatmap representing differential gene expression from sorted cell populations by FACS based on size and GFP(+/-) expression. (**E**) Dot plot of secreted, diagnostic, and vaccine antigens grouped by annotated cell type. Color indicates average gene expression per cluster. Circle diameter denotes proportion of cells in each cluster expressing the indicated gene. Presence of a blue box indicates whether the resultant protein from the gene is considered an abundant secretory antigen, a diagnostic antigen (Diagnos.) or whether the resulting protein has been investigated as a vaccine target (Vacc.). MS, muscle; MD, mesoderm; C, coelomocyte; S, secretory; CA, canal-associated; IB, inner body.

The online version of this article includes the following source data and figure supplement(s) for figure 3:

**Source data 1.** List of differentially expressed genes identified in the *B. malayi* microfilariae (mf) 'largest' cell population collected via fluorescent-activated cell sorting (FACS) and bulk RNA-seq.

**Source data 2.** List of differentially expressed genes in secretory cells (cluster 15).

**Source data 3.** List of differentially expressed genes identified in *C. elegans* large and GFP(+) cells collected via fluorescent-activated cell sorting (FACS) and bulk RNA-seq.

**Source data 4.** Comprehensive table of major antigens, vaccine targets, diagnostic markers, and notable secreted proteins.

**Figure supplement 1.** Representative fluorescent-activated cell sorting (FACS) gating scheme for collecting *B. malayi* single-cells based on size, granularity, and presence or absence of anti-*Wolbachia* IgG antibody.

**Figure supplement 2.** Representative fluorescent-activated cell sorting (FACS) gating scheme for collecting *C. elegans* strain BK36 cell populations.

**Figure supplement 3.** Distribution of gene transcripts across all clusters.

To assess whether the *C. elegans* excretory cell would have provided information relevant to unearthing the filarial secretory cell, we carried out pooled single-cell RNA-seq of GFP tagged (*vha-1p*::GFP) and sorted *C. elegans* excretory cells (*Figure 3D*, *Figure 3—source data 3*, and *Figure 3—figure supplement 2*). Established and new *C. elegans* excretory cell markers identified through this effort lack clear orthologs or display no meaningful overlap with highly or uniquely expressed orthologous markers found in the *B. malayi* secretory cell. This aligns with the distinct evolutionary history of the ES system in filariae and the parasite-specific functions the secretory cell is likely to perform in the host context.

## Abundantly-secreted parasite proteins have diverse tissue origins

We next investigated the origins of prominent molecules known to be secreted by blood-circulating mf and that are likely necessary for parasite immune evasion and survival. Studies have identified secreted proteins and antigens across the intra-mammalian life cycle stages (*Bennuru et al., 2011*; *Harischandra et al., 2018*; *Hewitson et al., 2008*; *Kaushal et al., 1982*; *Moreno and Geary, 2008*; *Zamanian et al., 2015*). We mined the literature for a comprehensive list of major antigens, vaccine targets, diagnostic markers, and other notable secreted proteins and probed the scRNA-seq data to identify their transcriptional origins (*Figure 3E* and *Figure 3—source data 4*). Transcripts encoding for secreted proteins exhibit variable patterns of expression across the body and very few are specifically localized to the secretory cell (cluster 15). Some prominent antigens that are expressed across many tissue types (*Bma-tpi-1* and *Bma-cpi-2*) have been localized to the ES apparatus via whole-mount antibody staining (*Moreno et al., 2010*), an observation that is likely reconciled by aggregation and secretion of these proteins through the ES apparatus. Additional broadly expressed antigens that are included as vaccine targets and diagnostic markers include *Bma-tpi-1*, *Bma-sxp-2*, and *Bm3837* (*Figure 3E* and *Figure 3—figure supplement 3*; *Krushna et al., 2009*; *Lalitha et al., 2002*; *Morris et al., 2013*).

We identified prominent antigens and diagnostic markers that exhibit much more tissue-restricted expression in both secretory and non-secretory cell types. *Bma-val-1*, *Bm97*, and chitinases (*Bma-cht-1.1* and *Bma-cht-1.2*) have strong and nearly undivided expression in the inner body (*Figure 3—figure supplement 3*). Muscle and undefined cell types also harbor an abundance of interesting transcripts (e.g., *Bma-unc-15*, *Bma-sxp-1*, *Bm7564*). Notable genes that showed strong or exclusive expression in the annotated secretory cells include microfilariae-specific serpin *Bma-srp-2* and

diagnostic antigens *Bm10631* (*BmR1*) (*Greene et al., 2022*) and *Bm14*, a Cys$_2$His$_2$ zinc finger (C2H2-ZF) transcription factor (*Figure 3E*).

## C2H2-ZF transcription factors are enriched in annotated secretory cells

We sought to further characterize transcripts that are specifically enriched in the annotated secretory cell. A greater fraction of proteins with transcripts enriched in the secretory cell contain signal peptides (43%) compared to other cell types (*Figure 4A* and *Figure 4—source data 1*). Signal peptides may not be required for release into the host environment as many proteins identified in ES studies do not possess this targeting signal (*Hewitson et al., 2008*). 41.8% of proteins with transcripts enriched in the secretory cell are predicted to contain transmembrane domains (*Figure 4B* and *Figure 4—source data 2*). Subpopulations of these putative membrane-spanning proteins may potentially serve as candidate hidden antigens that may evade recognition by the host immune system but may be highly accessible to vaccine-induced antibodies (Munn 1997).

We investigated the distribution of C2H2-ZF transcription factors in light of the high abundance and restriction of the diagnostic marker *Bm14* in the secretory cell cluster. We generated a phylogeny of *B. malayi* C2H2-ZF transcription factors and identified two expanded subgroups that are secretory cell specific and that lack one-to-one *C. elegans* orthologs (*Figure 4C*). Several of these transcription factors were also detected in the bulk RNA-seq data of FAC-sorted 'largest' cells. The C2H2-ZF family is a large, diverse family of proteins (*Kang and Kim, 2000*; *Razin et al., 2012*). Previous studies have reported an enrichment of this transcription factor class in mf-derived ES products and elevated expression of these genes in mf (*Bennuru et al., 2011*; *Choi et al., 2011*). The role of these proteins in ES cell types is poorly understood, but may involve governing gene regulatory events at the host–parasite interface or to help adapt mf for the stark environmental transition from the mammalian host to the mosquito vector.

## Mapping anthelmintic targets across the microfilarial body

Modulation of ES processes may explain the mechanism of action of essential anthelmintics (*Moreno et al., 2021*) and a deeper understanding of the cellular or physiological processes that regulate ES systems may identify new therapeutic strategies. Ion channels and structural proteins in the neuromuscular system are the primary targets of nearly all anthelmintics. To identify the distribution of the known and putative targets of existing and emerging anthelmintics within the microfilarial body, we mapped the cell-type locations of β-tubulins, cys-loop ligand gated ion channels (LGICs), and additional ionotropic receptors (*Figure 5A*). Benzimidazoles compromise nematode cell structural integrity by acting as microtubule inhibitors, but the specific β-tubulin target(s) are unknown in filariae despite the use of ABZ in MDA programs. The four β-tubulin *B. malayi* proteins exhibit broad expression across the mf body, with *Bma-btub-1* (*Bm4733*) most highly expressed compared to all other β-tubulins (*Figure 5A and B*). This β-tubulin also has the highest sequence similarity to the benzimidazole target in *C. elegans* (*ben-1*), and we hypothesize that *Bma-btub-1* is responsible for mediating the antifilarial action of ABZ. *Bma-tbb-4* and *Bma-mec-7* were minimally represented but found in muscle and neuronal cell types. The coexpression of β-tubulins in *C. elegans* cells provides redundancy that allows for normal growth in *Cel-ben-1* loss-of-function mutants (*Driscoll et al., 1989*). Single-cell coexpression of β-tubulins within individual *B. malayi* mf cells shows strong association between *Bma-tbb-4* and *Bma-btub-1* and a moderate association between *Bma-btub-1* and *Bma-btub-2* (*Figure 5C*), pointing to potential redundancy and compensatory drug response mechanisms.

Transient receptor potential (TRP) channels are gaining recognition as anthelmintic targets (*Park et al., 2019*) and the TRPC-like channel *Bma-trp-2* has recently been implicated as a target of DEC (*Verma et al., 2020*; *Williams et al., 2022*). We observed the expression of all 10 *B. malayi* TRP channels with at least one TRP channel subunit expressed in every cell type (*Figure 5A* and *Figure 5—figure supplement 1*). *Bma-trp-2* facilitates inward calcium currents upon activation and results in subsequent opening of the SLO-1 K$^+$ channel (*Verma et al., 2020*; *Williams et al., 2022*), the proposed target of the macrofilaricidal emodepside (*Kashyap et al., 2019*). The majority of *Bma-trp-2* and *Bma-slo-1* transcripts are found in body wall muscle (clusters 2 and 19), consistent with electrophysiological assays (*Kashyap et al., 2019*; *Verma et al., 2020*) and the paralytic effect of emodepside.

The glutamate-gated chloride channels (GluCls) are of special interest as the targets of the macrocyclic lactones and as regulators of ES processes. There is strong evidence that IVM acts on

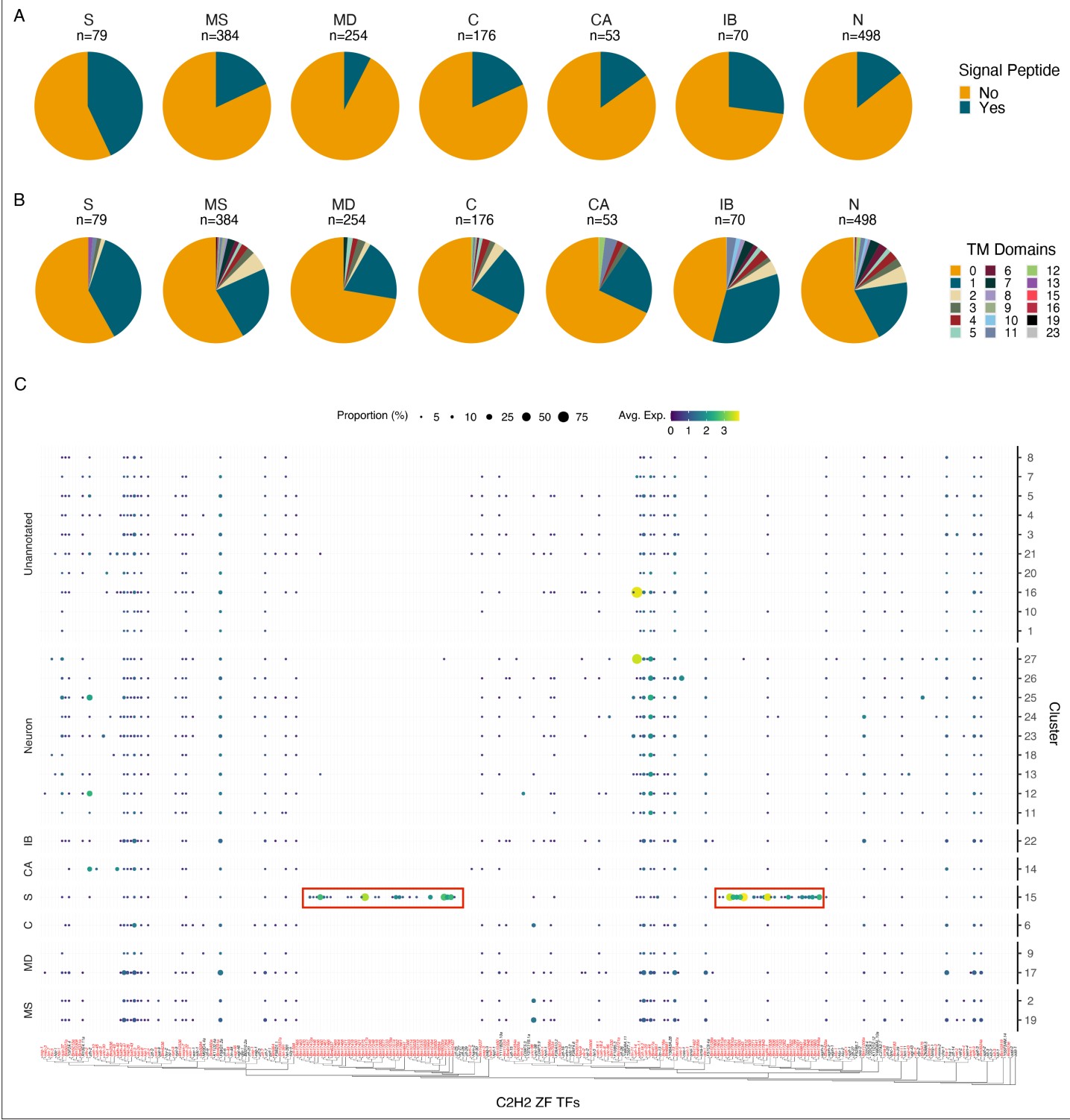

**Figure 4.** Characterization of the secretory cell indicates an enrichment in signal peptide containing proteins and C2H2 zinc finger transcription factors. (**A**) Quantification of signal peptide-containing sequences among all differentially expressed genes (DEGs) (p≤0.05) across major annotated cell types. Differentially expressed transcripts in the secretory cell show an enrichment for signal peptide sequences. S, secretory; MS, muscle; MD, mesoderm; C, coelomocyte; CA, canal-associated; IB, inner body; N, neurons. (**B**) Quantification of transmembrane domains of differentially expressed transcripts in the secretory cell. Secretory cells contain several genes with a single predicted transmembrane domain but very few proteins contain more than one domain. (**C**) Phylogenetic and expression analysis of C2H2 zinc finger transcription factors in *B. malayi* (red) based on orthologous C2H2-ZF TFs confirmed through direct evidence in *C. elegans* (black). Paralogous expansion (red boxes) of these transcription factors is observed with nearly

*Figure 4 continued on next page*

*Figure 4 continued*

exclusive expression in the secretory cell. A summary of Gene Ontology (GO) enrichment for DEGs of the secretory cell can be found in *Figure 4—figure supplement 1*.

The online version of this article includes the following source data and figure supplement(s) for figure 4:

**Source data 1.** Summary table of signal peptide sequence predictions in differentially expressed transcripts across major annotated cell types.

**Source data 2.** Summary of transmembrane domain prediction in differentially expressed transcripts across major annotated cell types.

**Figure supplement 1.** Gene Ontology (GO) term enrichment analysis for differentially expressed genes identified in the secretory cell.

muscle-expressed GluCls adjacent to the secretory apparatus leading to suppression of secretory cargo and host clearance of mf (*Harischandra et al., 2018*; *Li et al., 2014*; *Loghry et al., 2020*; *Moreno et al., 2010*). The *B. malayi* GluCl complement (*Bma-avr-14*, *Bma-glc-2*, *Bma-glc-3*, and *Bma-glc-4*) display expression in neuronal and muscle cell types (*Figure 5A and B*). *Bma-avr-14* is noted as an important subunit for IVM sensitivity (*Dent et al., 2000*), and the expression of *avr-14* in annotated muscle cell types is consistent with its expression in muscle surrounding the ES pore (*Moreno et al., 2010*). While functional *B. malayi* GluCl channels have been constituted in heterologous systems to determine IVM sensitivity (*Choudhary et al., 2022*; *Lamassiaude et al., 2022*), not all subunit compositions are amenable to expression and it is unclear what channel formations best reflect the native state (*Lamassiaude et al., 2022*). To improve predictions of native channel compositions, we examined the correlation of GluCl subunit expression in individual cells and identified *Bma-avr-14* and *Bma-glc-4* as the most commonly coexpressed GluCl subunits (*Figure 5C*). *Bma-avr-14* was also coexpressed with *Bma-glc-2*, a combination reported to be IVM-sensitive; however, *Bma-glc-2* was minimally expressed and was not coexpressed with any other GluCl subunit.

Nicotinic acetylcholine receptors (nAChR) are activated by levamisole (LEV) and other cholinergic anthelmintics. As expected, the subunits that comprise *L*-type and *P*-type nAChR channels are mainly restricted to the body wall muscle (*Figure 5A* and *Figure 5—figure supplement 1*). Single-cell coexpression analysis of nAChRs reveals that *Bma-acr-15* could be an essential and missing component of our understanding of nAChR composition and function in filarial nematodes (*Figure 5C*). These data enrich our understanding of tissue-specific targeting of the major anthelmintic classes and provide new cell-specific leads belonging to traditionally druggable receptor families.

## Measuring the effects of anthelmintics on the viability and transcriptional states of isolated cells

With a view of how anthelmintic targets are distributed across cells and tissues, we set out to investigate anthelmintic responses at single-cell resolution. To first validate flow cytometry as a cell viability quantification approach, single-cell suspensions from the same *B. malayi* dispersion reaction were exposed to DMSO, methanol (nonviable), or remained untreated (viable). Cell viability was successfully measured using DRAQ5 (nucleated objects) and Calcein Violet-AM, a shorter wavelength version of the Calcein-AM dye (*Figure 6A*). We also confirmed that cell health did not decline significantly after dissociation and incubation on ice in media for up to 8 hr and that DMSO at concentrations required for drug solubility did not compromise cell health (*Figure 6B*).

Next we examined the effects of three major anthelmintic classes (macrocyclic lactones, benzimidazoles, and nicotinic agonists) on parasite cell viability. Homogeneous cell preparations from independent parasite infection cohorts were split into parallel treatment conditions and cells were exposed to drugs for a 20 min incubation period prior to flow cytometry analysis. Dose–response curves (50 nM, 1 µM, 50 µM, 100 µM) were completed using IVM, LEV, and albendazole sulfoxide (AZS), the active metabolite of albendazole (*Figure 6C*). We did not observe a significant decrease in cell viability in any of the primary classes of drug at pharmacologically relevant concentrations; however, higher concentrations of IVM did produce a cytotoxic effect ($EC_{50}$ = 51 ± 5.16 µM) (*Figure 6D*).

While these anthelmintics did not exhibit strong cell-cidal activity, we reasoned they may differentially alter gene expression in various cell types. Single-cell RNA-sequencing was carried out using a homogeneous *B. malayi* mf cell suspension that was equally split with one half receiving short-term IVM exposure at a concentration that does not affect cell viability (1 µM for 20 min). Compared to the untreated control, muscle, neurons, and unannotated cell types revealed the most significant and abundant upregulation of genes in response to IVM exposure (*Figure 6E* and *Figure 6—source data*

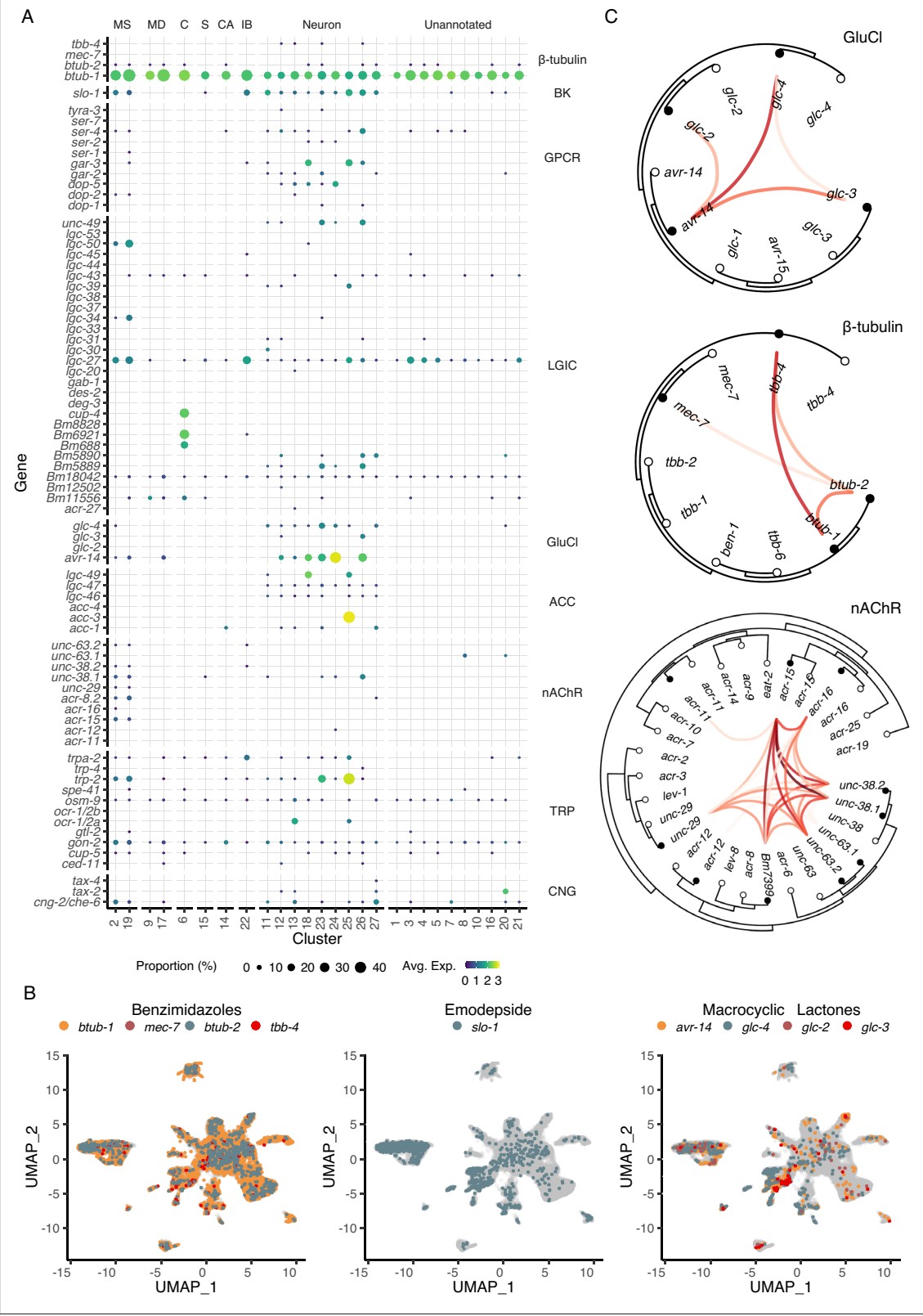

**Figure 5.** Distribution of putative anthelmintic targets and ligand-gated ion channel subunit colocalization. (**A**) Transcriptomic and gene expression profiles of major anthelmintic targets in microfilarial cell types. Targets include β-tubulins and cys-loop ligand-gated ion channel (LGIC) subunits corresponding to the following subfamilies: big potassium (BK), G protein-coupled receptor (GPCR), glutamate-gated chloride channel (GluCl), acetylcholine-gated channel (ACC), nicotinic acetylcholine receptor (nAChR), transient receptor potential channel (TRP), and cyclic nucleotide-gated

*Figure 5 continued on next page*

Figure 5 continued

ion channels (CNG). (**B**) Uniform Manifold Approximation and Projection (UMAP) depiction of cells expressing putative targets of benzimidazoles (β-tubulins), emodepside (*slo-1*), and macrocyclic lactones (GluCls). (**C**) Phylogenetic trees showing single-cell correlations between subunits belonging to GluCl and nAChR LGIC subfamilies and β-tubulins. Empty circles represent *C. elegans* and filled circles represent *B. malayi*. Red links indicate positive Pearson correlation coefficients calculated using the log normalized count values. S, secretory; MS, muscle; MD, mesoderm; C, coelomocyte; CA, canal-associated; IB, inner body.

The online version of this article includes the following figure supplement(s) for figure 5:

**Figure supplement 1.** Distribution of gene transcripts associated with anthelmintic targets across all cell clusters.

*1*). The greatest transcriptional shift occurs in muscle cells, which house the highest transcriptional abundance of GluCls (*Figure 5—figure supplement 1*). Differentially expressed genes of interest include serine protease inhibitor *Bma-srp-2*, a highly abundant ES product released from mf, which showed the greatest increase in expression in unannotated cluster 8 (Log$_2$FC > 2.8) but was also significantly upregulated in 9 of 27 clusters. The IVM response was additionally characterized by upregulation of genes previously identified in bulk RNA-seq studies (*Bm10661, Bm228, Bm9776, Bma-srp-2, Bm8514, Bma-unc-54.2, Bm33, Bm13260, Bma-rps-28*) (*Ballesteros et al., 2016*) and proteins with unassigned functions.

## Cell culture of dispersed mf suspensions provides avenue for single-cell functional analyses

Historically, cell culture models have revolutionized biological understanding and progress toward drug discovery and development in mammalian systems. The complex cyclodevelopmental life cycle of filarial worms and the absence of immortalized cell lines derived from any helminth species complicate the translation of these approaches to helminthology. While dispersions of embryonic *Brugia* cells have been performed, the captured cells do not differentiate in culture (*Higazi et al., 2004*), limiting the advantages of the system. To enable future studies of cell biology and drug interaction at single-cell resolution, we sought to establish a filarial nematode cell culture system for the longer-term maintenance of adherent and differentiated cell populations. To this end, *B. malayi* mf primary cell cultures were plated onto peanut-lectin coated chamber slides and imaged to assess cell adherence and health. Highly viable populations of adhered cells were observed after a 24 hr incubation in 10% FBS L-15 media (*Figure 7A*). We next plated primary cells onto peanut-lectin coated surfaces in microtiter plate format to monitor viability over a longer time frame using high-throughput imaging. Adhered cells remained viable at 96 hr as indicated by Calcein-AM fluorescence (*Figure 7B* and *Figure 7—figure supplement 1*). We assessed the mitotic activity of these cells using an EdU proliferation assay and determined these cells are not actively dividing in the current culture conditions (*Figure 7—figure supplement 2*). Further development and adaptation of this method will support transgenic manipulation and functional studies of differentiated *B. malayi* cell populations.

## Discussion

ES products released by parasitic nematodes into the host environment are recognized as key molecular mediators for the establishment and maintenance of infections. The secretory processes that underlie host immune evasion and modulation may serve as targets for existing (*Loghry et al., 2020*; *Moreno et al., 2010*) and future drugs. Despite this, our superficial knowledge of the structure and function of the secretory apparatus of human parasitic nematodes has limited our understanding of the molecular events that drive host–parasite interactions. To help address these knowledge gaps, we introduce the first single-cell transcriptomic atlas of a parasitic nematode and leverage this resource to resolve tissue and cell-specific gene expression patterns that allow for inferences about the origins of secretory products and antiparasitic mechanisms of action.

We developed a single-cell dissociation and RNA-seq protocol to generate a transcriptomic atlas of the blood-circulating microfilariae stage of the human filarial parasite *B. malayi*. Using marker genes specific for terminal and developing cell types, we annotated ~30% of the cells represented in our dataset as muscle and neuronally derived. Ten clusters in the dataset (61.5% of total cell count) remain unannotated. The use of a pre-larval developmental stage containing cells that have yet to terminally differentiate and a developmentally asynchronous mf population likely contributed to difficulties in

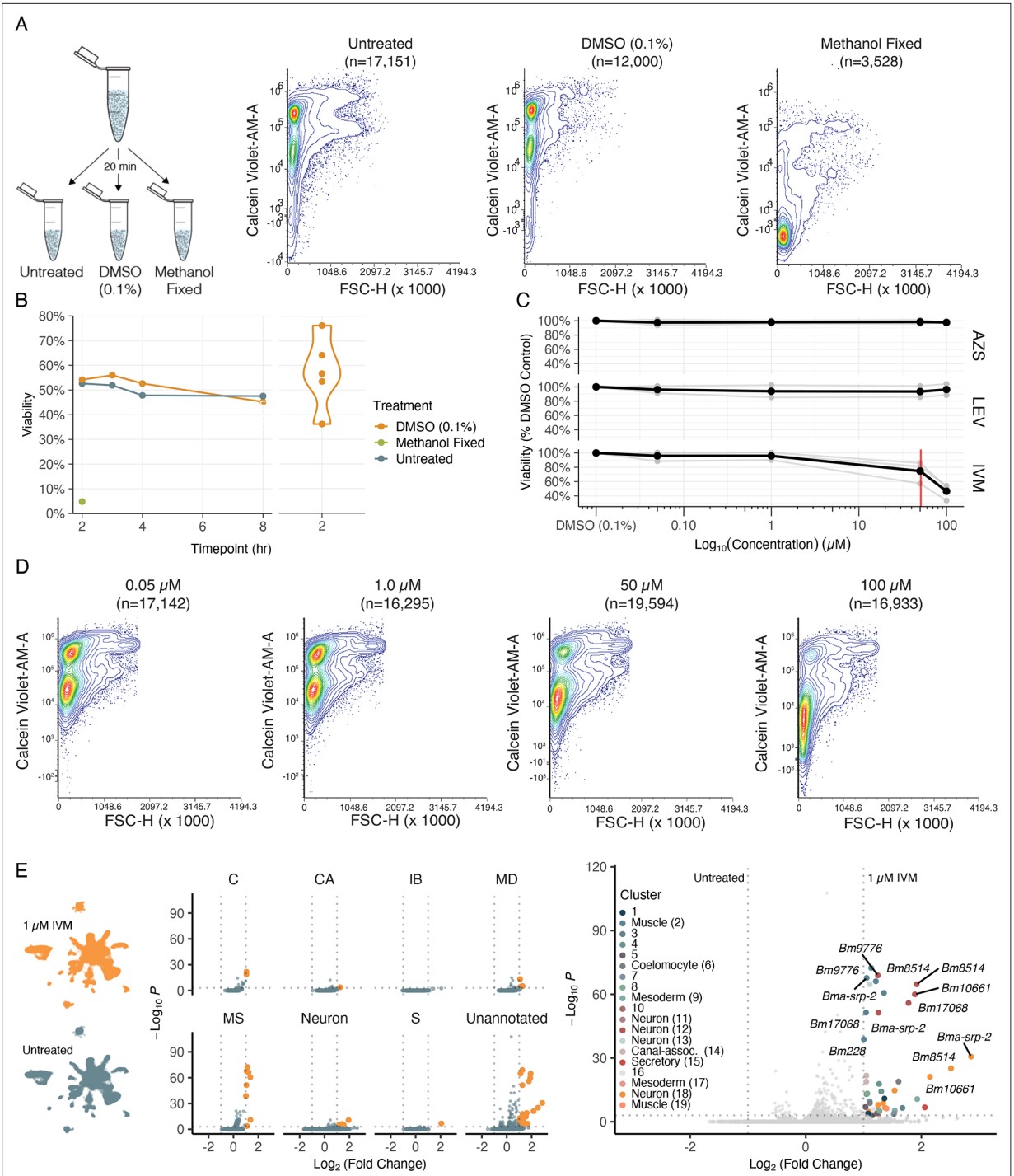

**Figure 6.** Cell viability and transcriptional shifts in response to anthelmintics. (**A**) Schematic and representative flow cytometry contour plots depicting Calcein Violet-AM (viable) fluorescence in control samples (untreated, 0.1% DMSO, and methanol fixed). Methanol fixed cells show no retention of Calcein Violet-AM indicated by decreased fluorescence in the violet channel. (**B**) Viability of dispersed cell suspensions incubated on ice for 8 hr. Significant variation in the percent viability of cell populations after dissociation was observed and attributed to the variation in mf health prior to dispersion as well as small nuances in pronase digestion. (**C**) Drug dose–response curves on dispersed single-cell suspensions at 50 nM, 1 μM, 50 μM, 100 μM, and a DMSO-only (0.1%) control. Viability was normalized to the percent DMSO control. AZS, albendazole sulfoxide; LEV, levamisole; IVM, ivermectin. Black line indicates the average of three biological replicates for IVM and two biological replicates for LEV and AZS. Red line indicates the EC$_{50}$ value for IVM at 51 ± 5.16 μM. (**D**) Representative flow cytometry contour plots of cell viability in response to IVM treatment. A decrease in Calcein Violet-AM fluorescence at 50 μM and nearly total cell death at 100 μM. (**E**) Left: single-cell transcriptomic response to IVM treatment (1 μM for 20 min, 1% DMSO). Uniform Manifold Approximation and Projection (UMAP) plot showing treated (top) and untreated (bottom) clustering of cell types. Middle:

*Figure 6 continued on next page*

*Figure 6 continued*

volcano plots depicting differentially expressed genes (DEGs) in treated and untreated groups by cell-type annotations. DEGs were calculated using Seurat's FindMarkers() function and comparing treated vs. untreated clusters. Orange color indicates upregulation in IVM-treated cells. Right: volcano plot of DEGs colored by cluster.

The online version of this article includes the following source data for figure 6:

**Source data 1.** Differential gene expression analysis between untreated and ivermectin (IVM) (1 μM)-treated single-cell suspensions.

comprehensive disentanglement and annotation of transcriptionally distinct cell clusters. The potential to annotate cells is further confounded by the unknown biological functions of genes that distinguish different clusters. These limitations can be observed in cluster 1, which represents nearly 27% of cells in the UMAP, and does not provide a distinct gene expression pattern indicative of a particular cell type. While it is possible this cluster is representative of unfiltered, low-quality cells comprised of ambient RNA, it is also plausible these cells represent low-expressing cells on a developmental trajectory or in a suspended state of development until the mf is transmitted to the mosquito host. Many unannotated clusters in the dataset may be resolved by deeper single-cell profiling and validation efforts. For instance, cluster 7 lowly expresses several *C. elegans* genes found in sensory neurons

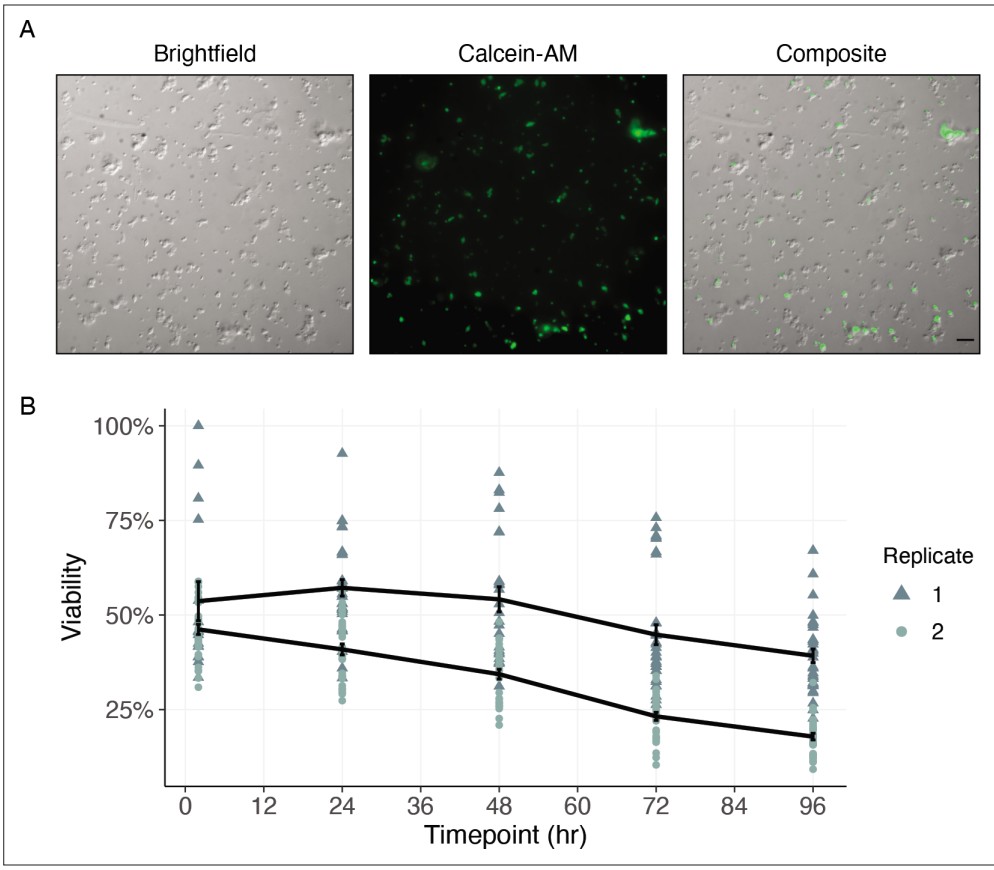

**Figure 7.** Cells of dispersed microfilaria are viable in culture 96 hr after dissociation. (**A**) Differential interference contrast (DIC) and fluorescence microscopy of cells from microfilarial stage *B. malayi* parasites on peanut-lectin-coated slides after 24 hr incubation in culture medium. The cell population contains a range of cell sizes including single-cells and clustered cells and a subset of the cell population have protrusions indicating a muscle or neuronal cell type. Calcein fluorescence (live) with varying intensities of brightness indicates a range of viability among the cell population. Scale bar = 20 μm. (**B**) Quantification of cell viability over a period of 96 hr using DRAQ5 and Calcein-AM viability dyes. Black line represents the mean and standard error of viability per replicate.

The online version of this article includes the following figure supplement(s) for figure 7:

**Figure supplement 1.** Representative images of cell viability over 96 hr.

**Figure supplement 2.** EdU cell proliferation assay of *B*.

related to cilium assembly and organization (*osm-1, dyf-11, ccep-290, ift-74, dyf-17, che-3, T12B3.1*) (*Inglis et al., 2006*) but lowly expresses only a subset of pan-neuronal markers (*Figure 2—figure supplement 3*).

Knowledge from *C. elegans* was useful in identifying rare cell populations such as coelomocytes or canal-associated cells, but alternative approaches were necessary for the identification of clade-diverged secretory cells. Pooled RNA-seq of sorted cell populations enriched for the morphological characteristics of the *B. malayi* secretory cell allowed for the annotation of secretory-like cells within the mf cell atlas. Surprisingly, prominent secreted proteins and antigens show different patterns of transcript distribution across cell populations, establishing that immunomodulatory ES products originate from different parasite tissues. For example, the diagnostic marker *BmR1* (*Bm10631*) is exclusively transcribed in the secretory cell while the vaccine immunogens *Bma-val-1* and *Bm97* are restricted to the inner body. It is possible that many of the proteins encoded by these transcripts share a common exit route from the worm through the ES apparatus (*Moreno et al., 2010*) or that many of these proteins are exposed to the host by dead or dying parasites. The latter possibility may warrant greater focus on targets that are actively transcribed in the secretory cell of viable parasites.

The inner body is a chitinase-enriched (*Bma-cht-1.1, Bma-cht-1.2*) intestinal tract precursor, and its breakdown leads to the release of chitinase and exsheathment within the mosquito (*Wu et al., 2008*). Exposure of mf to IVM prevents later exsheathment in the mosquito (*Rao et al., 1992*), which we speculate is consistent with suppression of ES function leading to the trapping of inner-body enzymes. This view of the mf ES apparatus as a secretory choke point in the parasite life cycle offers a control target relevant to both the arthropod and mammalian hosts. Later developmental stages in the parasite life cycle (L3 and adults) present additional orifices and interfaces for host interaction, necessitating other approaches to parse out the relationship between ES products and the ES apparatus in these stages.

The putative targets for mainstay anthelmintics are known, but their tissue distributions throughout the filarial worm body are unresolved. Further, the native subunit compositions of the ion channel drug targets of the macrocyclic lactones and nicotinic agonists are unknown. We leveraged these data to map the expression of known and putative antifilarial drug targets across cell types and to predict ion channel subunit associations from single-cell coexpression analysis. These predictions can guide the reconstitution and functional expression of pharmacologically relevant ion channels in heterologous studies of channel function and anthelmintic responses. We directly assessed the effects of anthelmintics on the viability of isolated cells, showing that the major anthelmintic classes do not elicit cell death at pharmacologically relevant concentrations. This suggests that the primary action of these drugs requires cell–cell connectivity. The action of IVM in *C. elegans* is innexin-dependent (*Dent et al., 2000*), underscoring the importance of gap junctions in the signaling cascade that results in the whole-organism drug effect. To capture early transcriptional events that drive IVM action in the mf stage, we identified core genes that are upregulated across isolated cell types in response to acute drug exposure. Future studies that compare the transcriptional trajectories of cells exposed to drug before and after dissociation may help resolve the role of cell–cell communication and various compensatory responses in anthelmintic mechanisms of action.

Finally, we developed a protocol for culturing cells derived from *B. malayi* microfilariae tissue dispersions. Efforts that improve our ability to manipulate these dispersed cell populations have the potential to expand our understanding of parasite cell biology and receptor-mediated responses to external drug and chemical stimuli. While there has been great progress in the genetic manipulation of filariae (*Higazi et al., 2002; Liu et al., 2020; Liu et al., 2018; Xu et al., 2011*), germline transgenesis has proven difficult and optimized protocols cannot overcome a two-host life cycle with a long parasite pre-patent period. Transfection of cultured *Brugia* mf primary cells with promoter-driven constructs and CRISPR reagents can help circumnavigate some existing limitations and facilitate the study of parasite protein function. Although primary cells are often difficult to transfect or transduce, optimization of both viral and nonviral approaches has helped overcome these challenges in a cell-type-specific manner (*Chong et al., 2021*). Collectively, the continued development of these resources and associated methods can help answer outstanding questions about the nematode parasite–host–drug interface.

## Materials and methods

### Data and code availability

All data and scripts used for data analysis and visualization are publicly available at https://github.com/zamanianlab/Bmsinglecell-ms, (copy archived at *Henthorn, 2023a*) and in a Zenodo repository (https://zenodo.org/record/7110316). Single-cell and FACS-pooled RNA-seq data has been deposited into NIH BioProjects PRJNA874113 and PRJNA874749. The processed data can also be explored at https://github.com/zamanianlab/BmSC_shiny (copy archived at *Henthorn, 2023b*) as a locally run RStudio Shiny application.

### Parasite and *C. elegans* strains

*Brugia* microfilariae (mf) were supplied by the NIH/NIAID Filariasis Research Reagent Resource Center (FR3) (*Michalski et al., 2011*). *B. malayi* mf were used for all experiments except for the imaging flow cytometry data where the anatomically equivalent *B. pahangi* species was used due to parasite availability at the time of the assay. Mf extracted from the jird peritoneal cavity were filtered and cultured in RPMI-1640 with L-glutamine (Sigma-Aldrich, St. Louis, MO) supplemented with 10% fetal bovine serum (FBS) (Thermo Fisher, Waltham, MA) and 50 μg/mL penicillin/streptomycin (P/S) (Thermo Fisher). All experiments used mf incubated at 37°C + 5% $CO_2$ for at least 1 hr prior to experimentation within 24 hr of host extraction, or stored at 4°C overnight in fresh media and processed within 48 hr of host extraction. *C. elegans* strain BK36 was acquired from the Caenorhabditis Genetics Center (CGC). *C. elegans* strains N2 and BK36 were maintained at 20°C on NGM plates seeded with *Escherichia coli* OP50. Worms were propagated by routine picking of L4 stage worms to seeded NGM plates.

### Microfilariae purification

*Brugia* mf were purified using a PD-10 desalting column (Cytiva, Marlborough, MA) to remove host cells, debris, embryos, and perished microfilariae as previously described (*Galal et al., 1989*) with minor changes. Briefly, the PD-10 column was equilibrated by passing 25 mL of RPMI-1640 with L-glutamine through the column. Mf were centrifuged at 2000 rpm for 10 min and the supernatant was drawn off, leaving 5 mL of peritoneal fluid and the mf pellet. Mf were resuspended, and the entirety of the suspension was transferred to the equilibrated PD-10 column. RPMI-1640 medium was added to the column in 5 mL increments, and the flowthrough was checked every 5 mL for the presence of mf and host peritoneal cells. Mf were collected when no host peritoneal cells were present in the flowthrough and until mf were no longer being eluted. Recovered mf were washed twice in RPMI-1640 by centrifugation, resuspended in RPMI-1640, and allowed to recover at 37°C for at least 1 hr prior to beginning a single-cell dissociation.

### Microfilariae single-cell dissociation

PD-10 column purified mf from the same jird parasite infection cohort were used as input for *Brugia* dispersions at ~1 million mf per reaction. The mf were aliquoted to a 1.5 mL microcentrifuge tube, pelleted, and resuspended in 2 mg/mL chitinase from *Streptomyces griseus* (Sigma-Aldrich) and incubated at 37°C with light agitation for 20 min. mf were washed once with 1 mL RPMI-1640 medium, and the chitinase-treated pellet was resuspended in SDS-DTT (200 mM DTT, 0.25% SDS, 20 mM HEPES, pH 8.0, 3% sucrose) diluted 1:4 in Leibovitz's L-15 medium without phenol red (Gibco, Waltham, MA) and an adjusted osmolality of 340 mOsm using 1 M sucrose. 200 μL of diluted SDS-DTT was added to the mf pellet and placed on a nutator for 6 min, where the mf are alive but not thrashing. 1 mL of L-15 medium was added to the tube and centrifuged at 16,000 rcf for 30 s at 4°C. The supernatant was removed, and the pellet was resuspended in 1 mL fresh L-15 medium. The mf were washed a total of five times or until the smell of SDS-DTT was no longer lingering. The pellet was resuspended in 100 μL pronase from *S. griseus* (VWR, Radnor, PA) at a concentration of 15 mg/mL in L-15 medium (340 mOsm). The reaction was continuously pipetted 100 times every 5 min either manually or by the use of the Repetitive Pipettor robot. The reaction was monitored by eye by checking small aliquots throughout the digestion. The reaction was stopped with ice-cold 1 mL L-15 + 10% FBS when most mf were broken open and single cells were clearly visible in the background (~30 min). The dispersed cells and remaining worms and debris was pelleted by centrifugation at 1000 rcf for 6 min at 4°C. The pellet was resuspended in 1 mL cold L-15 + 10% FBS and centrifuged briefly for 5 s at 1000 rcf to separate remaining large debris and single-cells. The top ~900 μL of cell suspension was drawn up

into a 1 mL syringe and pushed through a reusable syringe filter (Millipore, Burlington, MA) containing 7 µm mesh (Component Supply, Sparta, TN). The filtrates were combined and used in downstream applications. A standard hemocytometer was the most cost- and time-effective cell count estimation method with the caveat that differentiating cell versus debris was very difficult. Comparisons between the methods can be found in the supplemental material (*Figure 1—source data 1*).

## Imaging flow cytometry

Single-cell suspensions maintained on ice were incubated with DRAQ5 (500 nM) (BioLegend, San Diego, CA) and Calcein-AM (500 nM) (BioLegend) in L-15 + 10% FBS for a minimum of 20 min on ice prior to acquisition. An Amnis ImageStream Mark II Imaging Flow Cytometer equipped with a ×60 objective and a 2-laser (488 nm and 642 nm), 6-channel detection system was used for single-cell imaging acquisition. Samples were gated based on viability (DRAQ5+, Calcein-AM+). Analysis of images was completed using the Amnis Image Data Exploration and Analysis Software (IDEAS) and the integrated Feature Finder Wizard to distinguish viable single cells from the population based on a subset of hand-picked single cells as a training dataset.

## Anthelmintic drug dose responses

Single-cell suspensions generated from *B. malayi* microfilariae were subjected to drug treatment using IVM (Fisher), AZS (Fisher Scientific, Hampton, NH), and LEV (VWR). Biological replicates were taken from separate jird parasite infection cohorts. Drugs were suspended in DMSO (Santa Cruz Biotechnology, Dallas, TX) and added to cell suspensions at 50 nM, 1 µM, 50 µM, and 100 µM for 20 min on ice with a final DMSO concentration of 0.1%. At the completion of incubation, the cell suspensions were centrifuged at 1000 rcf for 6 min and as much supernatant as possible was removed without disrupting the pellet. Cells were resuspended in 300 µL of L-15 + 10% FBS containing DRAQ5 (2.5 µM) and Calcein Violet-AM (0.75 µM) and stored on ice until flow cytometry analysis. Treatment controls included untreated and DMSO only (0.1%) treated cell suspensions. Controls for spectral flow cytometry included L-15 + 10% FBS, unstained cells, and DRAQ5 (2.5 µM) and Calcein Violet-AM (0.75 µM) fluorescence minus one controls. Additional controls of L-15 + 100 µM IVM and unstained cells + 100 µM IVM were necessary to confirm the presence of IVM precipitation at high concentrations picked up by the spectral flow cytometer. Viability curves of single-cell suspensions over time were generated using cells from a *B. malayi* mf dissociation reaction (input of ~1 million mf) and split to create three treatment groups (untreated, 0.1% DMSO, and methanol fixed) with four samples each to cover four timepoints over 8 hr. Samples were stained with DRAQ5 and Calcein Violet-AM 30 min prior to analysis on the flow cytometer. Methanol fixed cell suspensions served as a cell death control and were prepared by adding 300 µL methanol (chilled to –20°C) to 100 µL cell suspension and incubated at 4°C for 30 min prior to staining and analysis. Samples were analyzed on a Cytek Aurora Spectral Cytometer equipped with five lasers (355 nm UV, 405 nm violet, 488 nm blue, 561 nm yellow-green, and 640 nm red). The laser light channels include a forward scatter and side scatter off of both the blue and violet lasers and has 64 fluorescence channel detectors. Selection of single-cell entities was completed using FSC-A vs. FSC-H followed by selection of cells that are DRAQ5(+) and Calcein Violet-AM(+). Gated cells with a Calcein Violet-AM intensity greater than 105 were considered Calcein Violet-AM(+) and viable. Percent viability was calculated by dividing the Calcein Violet-AM(+) population by the total DRAQ5(+) population, and the $EC_{50}$ was calculated using the R statistical software with the drc package (*Ritz et al., 2015*).

## 10x Genomics preparation and sequencing

A *B. malayi* mf single-cell suspension was split into two equal parts and one part was treated with 1 µM IVM in DMSO (1.0% final v/v) for 20 min on ice. At the end of incubation, the cells were centrifuged at 1000 rcf for 6 min at 4°C, the supernatant was removed, and the cells were resuspended in L-15 + 10% FBS. Cells were normalized to 1400 cells/µL as estimated by standard hemocytometer (INCYTO, Cheonan-si, South Korea) to target an input of 12,000 cells for capture on the 10x Genomics Chromium controller. A single technical and biological replicate was completed for each treatment. The 10x Genomics 3′ Single Cell RNA-seq protocol (v3.1 NextGEM User Guide Rev.D) was followed to generate the gel bead-in-emulsions (GEMs) and post cDNA amplification cleanup. The cDNA library was prepared using the Chromium Dual Index TT primers and the Single Cell 3′ v3.1 Reagents and

user guide. The library was sequenced on an Illumina NovaSeq6000 to generate ~550M reads across control and drug-treated samples (2 × 50 bp on S1 flow cell).

## scRNA-seq mapping

The ivermectin-treated and untreated *B. malayi* mf scRNA-seq data were mapped and processed independently. Single-cell RNA-seq data were mapped to the *B. malayi* reference genome (Wormbase, WS285) using 10x Genomics Cell Ranger 5.0.1 analysis pipeline. The 10x Genomics 3′ Single Cell RNA-seq protocol is strongly biased toward the 3′ regions of mRNA that are not well annotated in *B. malayi,* thus requiring a modified reference genome to improve the mapping rates of the scRNA-seq data. The 3′ UTRs were artificially extended by 50, 100, 150, 200, 250, 300, 400, and 500 bp as previously described with modifications (*Taylor et al., 2021*). Briefly, the scRNA-seq data was mapped using the Cell Ranger pipeline for each iteration, and the total reads for each gene in every iteration were summarized across all of the cells. The gene counts for each extension were normalized to the gene count of the 500 bp extension, and the optimal extension length for each gene was determined by identifying the extension that provided at least a 90% increase in mapping rate. The *B. malayi* scRNA-seq raw data was mapped one final time using the newly generated reference genome containing the optimal 3′ UTR extension length for each gene. The Cell Ranger filtered output drastically underreported the estimated cell count and is not optimized for small, low-expressing cells. Therefore, the raw Cell Ranger output was used for filtration and preprocessing.

## scRNA-seq bioinformatics analysis

The raw Cell Ranger barcode matrix output for each scRNA-seq dataset (untreated and +IVM treated) was filtered to remove empty droplets using the cluster-based R package scCB2 (*Ni et al., 2020*). Using EmptyDrops (ED) as a scaffold, scCB2 increases the power to distinguish real cells from background empty droplets as well as low-expressing, small cells by pooling low-count barcodes with similar gene expression patterns. The cluster is then compared against the estimated background population to identify irrelevant barcodes. This method of empty droplet filtration allows for the identification of small, low-expressing cells that are prevalent in the *B. malayi* mf single-cell dispersions. To facilitate this testing, scCB2's C2FindCell() function was used and droplets with 100 or fewer total counts were used to estimate the background distribution (lower = 100). A false discovery rate (FDR) was set to 0.01. The resulting cell matrix was converted to a Seurat object, and genes detected in less than three cells were omitted. The filtered cell population was then used as input into the R package SoupX (*Young and Behjati, 2020*) to estimate the contribution of cell-free mRNA contamination captured in each droplet. Because SoupX can provide a better estimate of contamination when cluster information is supplied, the scCB2-filtered datasets were taken through the Seurat pipeline (*Hao et al., 2021*) to identify clusters. Briefly, each dataset was log normalized and variable features were selected using the 'vst' method and 2000 features. Principal component analysis (PCA) was completed using 50 dimensions, which were used for clustering and dimension reduction via UMAP. The raw and clustered datasets were used as input to SoupX with cells containing 0–25 total UMI counts used to estimate the 'soup.' A lower UMI count to estimate the 'soup' was used to avoid unintentionally including lower expressing neuronal cells. The estimated contamination fraction for the untreated and treated datasets were calculated to be approximately 0.13 and 0.10, respectively, and the count matrix was adjusted accordingly using the adjustCounts() function. The ambient mRNA contamination was removed and additional filters were applied on the remaining cell population. Cells with high mitochondrial representation (≥10% of all UMIs per cell) were identified using the mitochondrial markers *Bm5157, Bma-nduo-4,* and *Bma-ctc-1* (*Qing et al., 2021*). Cells were removed from the population if a single gene represented ≥15% of the total cell transcripts, had >2500 total genes expressed, and >1800 total UMIs. These cutoffs were determined by the distributions and to omit potential doublets. The remaining count matrices for the untreated and treated datasets contained >10,000 genes with 21,131 and 25,490 cells, respectively. Seurat 4.1.1 was used for downstream integration and normalization. First, each matrix was log normalized and the top 2000 variable genes were identified using the FindVariableFeatures() functions using the 'vst' selection method. Variable genes were then used to integrate the control and treated datasets using the FindIntegrationAnchors() and IntegrateData() functions with default parameters. PCA (dims = 50) for dimension reduction was completed for clustering and visualization of the combined datasets. Differentially

expressed genes between clusters were completed using the FindConservedMarkers() function in Seurat, which uses a Wilcoxon rank-sum test and Bonferroni p-value correction to account for false positives. Genes with an adjusted p-value>0.05 were considered significant. Differentially expressed genes in response to IVM treatment were identified using Seurat's FindMarkers() function with a logfc.threshold=0.25. Treated and untreated cells were compared by cluster assignment.

## Pseudobulk analysis

Pseudobulk analysis was completed using the single-cell dataset generated from *B. malayi* mf. The raw read counts from the gene-by-cell expression matrix were aggregated for each cluster and treatment assignment. Each cluster and treatment combination was handled as an individual sample (1_tBM = 1 sample). The size of the aggregated read counts for each sample was TMM normalized using edgeR (v3.38.4; *Robinson et al., 2010*). Genes with less than 10 aggregated read counts across all samples were removed from the dataset. The pseudobulk gene expression matrix was $\log_2(x+1)$ and the resulting counts were median-centered. Genes and samples were hierarchically clustered in R using the ClassDiscovery package (v3.4.0; *Coombes, 2021*) using an uncentered Pearson correlation to calculate the distance matrix and complete-linkage for clustering. The gene expression heatmap was generated using pheatmap (v1.0.12; *Kolde, 2018*) where color indicates the level of expression (yellow = upregulated; blue = downregulated).

## Phylogenetic analyses

*B. malayi* transcription factors were identified using a reciprocal blastp and HMM approach using *C. elegans* transcription factors obtained from the CIS-BP database (v.2.0; Weirauch et al. 2014) as input. Only transcription factors that were supported by direct experimental evidence were considered. Multiple sequence alignment was completed using MAFFT (v.7) with an iterative refinement approach (E-INS-i) and poorly aligned regions were removed using trimAI (v.1.2; *Capella-Gutiérrez et al., 2009*). A maximum likelihood phylogenetic tree was built using iqtree (v.2.2.0; *Minh et al., 2020*) and the ultrafast bootstrap approximation (UFBoot) method requiring a minimum of 1000 replicates (*Hoang et al., 2018*). The resulting tree was visualized in RStudio using the ggtree R package (v.3.4.4; *Yu et al., 2017*). This method was also used to identify orthologous and paralogous C2H2 zinc finger transcription factors in *B. malayi*.

## *C. elegans* single-cell isolation

Chunks (~5 mm²) of media from 4-day-old plates were excised by flame-sterilized spatula, transferred to 5–10 new 10 cm NGM plates, and incubated at 20°C. At 102–120 hr post chunking, worms were washed off plates with 15 mL M9, pelleted (~1100 rpm/5 s), and exposed to 6 mL of freshly prepared bleaching solution (20% NaOCl, 0.5 M NaOH in ddH₂O) with mild rotation until partially degraded bodies were observed (6–8 min) under Stemi 508 stereo-microscope. Eggs were pelleted (~1100 rpm/5 s) and washed 3 × in 15 mL M9 media. After the final wash, embryos were resuspended in 2–4 mL M9 and counted in 5 × 5 μL droplets on a glass slide under a Zeiss Stemi 508 stereo-microscope. 2000–4000 embryos were plated in 125–150 μL M9 on 20 (N2) or 55 (BK36) 10 cm NGM plates seeded with OP50 and kept at 20°C for 46.5 hr prior to single-cell isolation. Single cells were isolated from bleach synchronized *C. elegans* N2 and BK36 L4 stage worms using an established protocol with modifications (*Kaletsky et al., 2016*). Worms were transferred in ~30 mL M9 media to 3 × 1.5 mL centrifuge tubes by sequential pelleting by microfuge followed by washing 5× in 1.5 mL M9. Worms were washed once in 500 μL lysis buffer (200 mM DTT, 0.25% SDS, 20 mM HEPES pH 8.0, 3% sucrose) then incubated for a further 8 min in lysis buffer with continuous rotation. Digested worms were washed 5 × in M9 then subject to 500 μL pronase 20 mg/mL from *S. griseus* in L-15 for 14–15 min at room temperature (RT) with semi-continuous pipetting. Pronase treatment was stopped with FBS (3% final v/v). Large debris was pelleted by microfuge, and cells were filtered (7 μm nylon mesh), stained with 0.5 μg/mL DAPI, and kept on ice. Throughout the protocol, samples were checked under stereo and fluorescent microscopes to determine the release and integrity of cells.

## FACS pooled RNA-seq

FACS was performed using a FACS Aria II (Becton Dickinson) for both *C. elegans* and *B. malayi* cell suspensions. *C. elegans* BK36 (*Mattingly and Buechner, 2011*) single-cell suspensions

generated from L4 stage larvae were sorted directly into TRIzol LS (Invitrogen, Waltham, MA) in 1.5 mL microfuge tubes. N2 strain cells were used as reference. Gates included dead cells (DAPI+, 4',6-diamidino-2-phenylindole, 0.5 µg/mL), GFP+ cells (green fluorescent protein), and cell size by aspect ratios, respectively (*Figure 3—figure supplement 2*). Total RNA was purified using Direct-zol RNA Microprep kit (Zymo Research, Irvine, CA), eluted in a minimal volume of nuclease-free water and flash frozen in liquid nitrogen or dry ice and stored at –80°C. RNA integrity was determined by Agilent RNA 6000 Pico Kit (Agilent, Santa Clara, CA) on a 2100 Bioanalyzer (Agilent) before library preparation. Total RNA was converted to double-stranded cDNA and amplified using SMART-Seq v4 (Takara, Kusatsu, Shiga, Japan). Full length cDNA was quantified by 2100 Bioanalyzer (Agilent). 150 pg of amplified cDNA was tagmented and index-amplified using Nextera XT adapters (Illumina). Library quantity was assessed by Qubit HS DNA and quality assessed by 2100 Bioanalyzer (Agilent). Libraries were then balanced and 2 × 75 bp sequencing was carried on the Illumina MiSeq.

For *B. malayi* single-cell suspensions, gates were set to collect live single cells (DRAQ5+, DAPI-) based on size (small, large, largest), granularity, and presence or absence of *Wolbachia* antibody (*Figure 3—figure supplement 1*). Presence of *Wolbachia* identified by fluorescent conjugated (Lightning-Link R-Phycoerythrin Conjugation Kit #703-0030) *Wolbachia* surface protein *wBm*0432/GenPept: WP_011256630 mouse monoclonal antibody (BEI Resources, Manassas, VA). Cells were sorted into 750 uL TRIzol LS, and libraries were prepared using the NEBNext Single Cell/Low Input RNA Library Prep Kit for Illumina (NEB, version 3.0, #E6420L) and NEBNext Multiplex Oligos for Illumina (Index Primers Set 1, NEB, #E6440G), followed by 2 × 150 bp sequencing on the Illumina MiSeq. Single technical replicates for *C. elegans* strains and *B. malayi* mf dispersions were completed.

## Secretory cell characterization

Differentially expressed genes for each major annotated *B. malayi* cell type were identified using the FindConservedMarkers() from the Seurat R package with an adjusted p-value≤0.05. Gene Ontology (GO) IDs for annotated *B. malayi* genes were acquired from WormBase ParaSite (*Howe et al., 2017*) using the biomaRt R package (v.2.52.0; *Smedley et al., 2009*), and analysis of enriched secretory cell transcripts was completed using the topGO R package (v.2.48.0; *Alexa and Rahnenfuhrer, 2010*). Significantly enriched GO terms were identified using the topGO's Fisher's exact test (p≤0.05). The protein sequences for differentially expressed secretory cell genes were obtained from WormBase and used as input into SignalP 6.0 (*Teufel et al., 2022*) to computationally predict signal peptide sequences. The 'slow' model and Organism option 'Other' were used as settings. Transmembrane domains were predicted and quantified using the same list of differentially expressed genes for each major cell type as input to HMMTOP 2.1 (*Tusnády and Simon, 2001*).

## Microfilariae cell culture

Single-cell suspensions for the purpose of cell culture were prepared as previously described with the exception of the pronase digestion step. After the initial 30 min incubation with mechanical digestion, the suspension was briefly centrifuged at 1000 rcf to pellet undigested worms. The supernatant was removed and supplemented with 1 mL L-15 + 10% FBS and placed on ice to inhibit further pronase digestion on released single cells. The worm pellet was resuspended in an additional 100 µL of pronase (15 mg/mL) for continued digestion for 30 min with manual pipetting to release cells from worms in early stages of digestion. At the end of pronase digestion, all microcentrifuge tubes were centrifuged at 1000 rcf for 6 min at 4°C to pellet undigested material and single cells. The supernatant was replaced with L-15 + 10% FBS, tubes were briefly centrifuged (~5 s), and the supernatant was filtered through 7 µm mesh. Peanut lectin-coated chamber slides or plates were prepared by incubating peanut lectin (0.5 mg/mL) (Sigma-Aldrich) suspended in water in chamber slide wells for 30 min, removed, and allowed to dry while being UV treated for 2 hr for sterilization. Filtered cell suspensions were concentrated prior to plating on peanut lectin-coated chamber slides or plates by centrifugation and resuspension in a smaller volume. Cells were seeded and incubated in a humid chamber at 37°C + 5% $CO_2$. Viability was visualized using live/nucleated cell staining with DRAQ5, DAPI, or Calcein-AM dyes. Two biological replicates generated from microfilariae obtained from different infected jirds were completed with at least two technical replicates from the same single-cell suspension dissociation reaction.

## Cell proliferation assay

*B. malayi* cell suspensions were assessed for proliferative activity using the thymidine analog EdU (5-ethynyl-2′-deoxyuridine) and the Click-iT EdU Cell Proliferation Kit for Imaging with the Alexa Fluor 488 dye (Thermo Fisher). Cells dissociated from *B. malayi* mf were seeded onto peanut-lectin coated, UV-treated Nunc Lab-Tek II 8-well chamber slides (Thermo Fisher) at a density of 30k cells/chamber in L-15 + 10% FBS + P/S. Cells were allowed to adhere overnight at 37°C + 5% $CO_2$ prior to incubating in EdU at 0 μM, 0.1 μM, 1.0 μM, 10 μM, and 50 μM concentrations. The CHO-K1 + Gα16 cell line was used as a positive proliferation control to confirm the EdU Click-chemistry reaction was functioning and was incubated with 0 μM and 10 μM EdU. Cells were incubated with EdU for 48 hr prior to detection of EdU incorporation using the kit protocol for cell fixation, permeabilization, EdU detection, and staining with Hoescht 33324 prior to imaging. Cells were imaged at 10× and 40× for CHO-K1 + Gα16 and *B. malayi* mf-derived cells, respectively.

## Microscopy

Differential interference contrast (DIC) imaging was completed using an upright Zeiss Axio Imager. D1 microscope equipped with a ×100 oil objective (Plan-Neofluoar ×100, NA = 1.3, Zeiss). All other brightfield and fluorescence microscopy used an ImageXpress Nano Automated Imaging System (Molecular Devices) with ×10 and ×40 air objectives.

## Integrating model nematode datasets

In an effort to identify cell types within the *B. malayi* scRNA-seq dataset, Scanorama was used as an integration tool with publicly available *C. elegans* scRNA-seq datasets based on one-to-one orthologs (*Hie et al., 2019*). The pre-processed untreated *B. malayi* dataset was reduced to only genes that have a one-to-one ortholog in *C. elegans* followed by replacement of the *B. malayi* gene IDs to *C. elegans* gene IDs using Seurat. The Seurat object was then exported as an Ann Data object for input into Scanorama. The L2 larval stage specific dataset (*Ben-David et al., 2020*) and the most notable L4 larval stage dataset (CeNGEN) (*Taylor et al., 2021*) were integrated with the subsetted *B. malayi* dataset with batch integration and plotted using ScanPy (*Wolf et al., 2018*).

## Repetitive pipettor (ReP)

The ReP is an automated, 3D-printed assembly that uses a DC motor (ServoCity, Winfield, KS), Raspberry Pi (rpi) Nano (CanaKit, North Vancouver, Canada), and an L298N motor controller (SunFounder, Shenzhen, China) to aspirate and dispense a p200 Eppendorf Research Plus pipette (Eppendorf, Hamburg, Germany) at any given rate, frequency, and volume. The structure was designed and modeled in Solidworks 2020 (https://www.solidworks.com/). The design includes six independent parts that were converted to STL files in Solidworks and sliced in Ultimaker Cura (https://github.com/Ultimaker/Cura) using the default printing settings of 0.2 mm layer height, 20% infill, and support material checked. The flat face of each part was positioned in contact with the build plate to reduce support material and refine rounded surfaces. Parts were printed using polylactic acid plastic filament and the support material was removed prior to assembly. Four pins were soldered to the rpi at pin locations 16, 18, 20, and 22. The rpi was subsequently glued to the 3D-printed structure at the four corner supports such that the SD card was facing upwards, and the USB and HDMI ports were facing outwards. The LN298 motor controller was glued to the opposite side of the structure such that the heat sink was facing downward. For the rpi, ports 16, 18, and 22 were general purpose input/output pins and were connected to the IN1, IN2, and ENA ports of the motor controller, respectively. Port 20 was a ground inserted at the GND port of the motor controller. The 12 V power supply had a positive and negative lead inserted into the motor controller at the +12 V and GND ports, respectively. The OUT1 and OUT2 ports on the motor controller were finally wired and soldered to the motor terminals. All code was written in Python, and the rpi was controlled remotely via SSH tunnels. All part files, protocol codes, and in-depth fabrication instructions can be found at https://github.com/emmakn/ReP, (copy archived at *Neumann, 2021*).

## Acknowledgements

The authors thank the University of Wisconsin Carbone Cancer Center (UWCCC) Flow Lab and the Cancer Center Support Grant P30 CA014520 for assistance with flow cytometry. The UWCCCC Cytek Aurora Spectral Cytometer was made available through a NIH Shared Instrumentation Grant (Grant #: 1S10OD025225-01, Project Title: BD FACSymphony High Parameter Flow Cytometer). The authors would also like to thank the University of Wisconsin Biotechnology Center Gene Expression Center for facilitating single-cell and low-input RNA-sequencing. Brugia life cycle stages were obtained through the NIH/NIAID Filarial Research Reagent Resource Center (FR3), morphological voucher specimens are stored at the Harold W Manter Museum at University of Nebraska, accession numbers P2021-2032. *C elegans* strains were provided by the CGC, which is funded by the NIH Office of Research Infrastructure Programs (P40 OD010440). Finally, the authors would like to thank Gabi Munoz for help optimizing single-cell suspensions and members of the Zamanian lab for constructive comments on the manuscript. This work was supported by the National Institutes of Health NIAID grant R01 AI151171 to MZ and the Wisconsin Alumni Research Foundation (WARF).

## Additional information

### Funding

| Funder | Grant reference number | Author |
| --- | --- | --- |
| National Institutes of Health | R01 AI151171 | Mostafa Zamanian |
| NIH Office of Research Infrastructure Programs | P40 OD010440 | Mostafa Zamanian |

The funders had no role in study design, data collection and interpretation, or the decision to submit the work for publication.

### Author contributions

Clair R Henthorn, Conceptualization, Data curation, Software, Formal analysis, Validation, Investigation, Visualization, Methodology, Writing - original draft, Writing - review and editing; Paul M Airs, Data curation, Investigation, Methodology, Writing - review and editing; Emma K Neumann, Methodology; Mostafa Zamanian, Conceptualization, Resources, Supervision, Funding acquisition, Validation, Investigation, Visualization, Methodology, Writing - original draft, Project administration, Writing - review and editing

### Author ORCIDs

Clair R Henthorn http://orcid.org/0000-0002-0224-7336
Paul M Airs http://orcid.org/0000-0003-0582-006X
Mostafa Zamanian http://orcid.org/0000-0001-9233-1760

### Decision letter and Author response

Decision letter https://doi.org/10.7554/eLife.83100.sa1
Author response https://doi.org/10.7554/eLife.83100.sa2

## Additional files

### Supplementary files

• MDAR checklist

### Data availability

All data and scripts used for data analysis and visualization are publicly available at https://github.com/zamanianlab/Bmsinglecell-ms, (copy archived at *Henthorn, 2023a*). Single-cell and FACS-pooled RNA-seq data has been deposited into NIH BioProjects PRJNA874113 and PRJNA874749.

The following datasets were generated:

| Author(s) | Year | Dataset title | Dataset URL | Database and Identifier |
| --- | --- | --- | --- | --- |
| Henthorn CR, Zamanian M | 2022 | Brugia malayi single-cell sequencing | https://www.ncbi.nlm.nih.gov/bioproject/PRJNA874113 | NCBI BioProject, PRJNA874113 |
| Henthorn CR, Zamanian M | 2022 | Nematode FACS-pooled RNA-seq | https://www.ncbi.nlm.nih.gov/bioproject/?term=PRJNA874749 | NCBI BioProject, PRJNA874749 |

The following previously published dataset was used:

| Author(s) | Year | Dataset title | Dataset URL | Database and Identifier |
| --- | --- | --- | --- | --- |
| Reaves BJ | 2017 | The effect on anthelmintic drugs on Brugia malayi gene expression in vivo | https://www.ncbi.nlm.nih.gov/bioproject/PRJNA388112 | NCBI BioProject, PRJNA388112 |

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
