## [Editor Report]

In this important study, the authors generate and analyse single-cell datasets for the human parasitic nematode Brugia malayi. The new resource has the potential to uncover new details of the biology of secretory systems in this filarial nematode. The analyses are of high quality but some concerns remain about the quality of parts of the data due to difficulties inherent in obtaining and preparing samples from this parasite. The new resource will be of broad interest to parasitologists and nematode biologists and has the potential to accelerate research in the search of new anthelmintics and vaccines.

---

## [Decision Letter]

**Decision letter after peer review:**

Thank you for submitting your article "Resolving the origins of secretory products and anthelmintic responses in a human parasitic nematode at single-cell resolution" for consideration by *eLife*. Your article has been reviewed by 3 peer reviewers, one of whom is a member of our Board of Reviewing Editors, and the evaluation has been overseen by Piali Sengupta as the Senior Editor. The following individual involved in review of your submission has agreed to reveal their identity: Michael Povelones (Reviewer #2).

Essential revisions:

1) The authors use single-cell transcriptomics without FACS, which is problematic. The main issue is that single-cell transcriptomic methods are so powerful that even without FACS meaningful biological cell type identities emerge. However, the authors go further to claim that secreted products come from a variety of cell types, and this is something that could come from the lack of FACS and a general background signal. The authors will need to validate the quality of their dataset despite the lack of FACS or provide alternative sources of information, be it a new dataset or other validation experiments. Alternatively, they need to tone down the claims about broad unspecific expression and critically discuss the caveats above. The authors will need to make a better effort in explaining the quality (or lack thereof) of their dataset, for the readers to be able to interpret the data.

2) If the authors are able to address the concerns about the quality of the dataset or generate new (e.g., FACS sorted) data, they should elaborate on the interpretation of their single cell data explore the new resource in more detail. For example, a more detailed bioinformatics analysis of the secretory products would provide added value to the paper. However, any further interpretations made from their raw data, without assessing or readdressing the quality of the dataset, are not recommended due to concerns regarding the quality of the data.

*Reviewer #1 (Recommendations for the authors):*

"neuromuscular and other cell types" – there are no neuromuscular cell types (only neuromuscular system), would be better to write neuronal and muscular cell types

The authors "mined the literature for a comprehensive list of major antigens, vaccine targets, diagnostic markers, and other notable secreted proteins"

It would be useful to provide this list with the references as a source data file

The authors give an EC50 value of 51 μM for IVM treatment. Given the relatively low number of concentrations tested and replicates, this seems too precise. Either they should also provide the error of the estimate, or only say that at 50 μM approximately half of the cells died.

Figure 1—figure supplement 2 – check figure, split across two pages, text should go into figure legend

What is the difference between Figure 4A and Figure 4 —figure supplement 1? "Transcript abundance of anthelmintic targets across cell clusters" it seems to be the same data plotted differently. This should be clearly explained (proportion vs total reads).

In general, the figure legends for the figure supplements and source data are minimal. These should be expanded to provide more explanation.

*Reviewer #2 (Recommendations for the authors):*

Given the lack of cell lines for detailed functional studies, the ability to culture dissociated cells is exciting. The authors studied the viability in culture of the dissociated cells. Do any of the dissociated cells divide? This can be an important prerequisite for some transformation protocols. Have attempts to transfected them been made? They have been cultured on surface treated with a lectin and standard culture surfaces. Were differences seen between those? Are the secretory cells present in the culture and maintain viability to 42 h? What happens beyond 42 h?

The fact that immunogenic proteins have a broad cell distribution in the mf is interesting and as noted, raises important questions about how these proteins are exposed to the host. A greater discussion of the immunogenic proteins is warranted. For example, could some immunogenic proteins be exposed to the host by dead or dying worms? Is there an argument to be made for focusing more exclusively on those proteins that are actively transcribed by the secretory cell. Perhaps not, but expanding on this topic would be helpful.

Despite the impressive effort in generating the data, they are mostly descriptive. Given the novelty and potential value of these data and methods to the community, this is ok, but clearer discussion on the potential for functional studies is needed. In its current form, the functional potential is too vague and in places reads more like an end, rather than a beginning. For example, one of the more exciting findings is the ability to capture the single secretory cell based on size and physical properties. A potential application could be to sequence transcripts this cell to a greater depth or perform proteomics on these cells to provide a more comprehensive repertoire of its secreted products. Or to follow it developmentally? Or to culture it?

This representation is used in several figures and it is sometimes unclear what is important. Consider adding boxes or some other way to highlight important trends in the dot plots. Also, for each dot plot, please make sure there are column label abbreviation keys in the figure legend.

In panel 1A, should there be mf in the illustration of the filtration column? The legend for panel 1B indicates that this is the pre and post PD10 filtration. In this case, it is clear there are fewer particles. What are these particles? There is no scale on this. Presumably there should be mf present and they would remain at the same concentration, but they are not visible? Panel 1C needs scale indicated as well even though the mf are clear.

Panel 2B is largely redundant with 2A. As stated above, I think text and number labels with circles around clusters would improve the clarity of 2A and make 2B unnecessary.

Legend for Figure 3F should include a legend for the column labels

Panel 3G could be streamlined. I would consider a more compact simpler representation of the data, in line with how it is discussed in the paper. The larger format as it stands now could be moved into the supplement. The number or specific members of the C2H2 Zn finger family members is not discussed in the text.

In Panel 5A, are only cells >10^5 in the violet channel considered viable? Either indicate in the text and/or on the graph what the cutoff is since the methanol fixed cells have signal just below 10^5 overlapping where there is a second lower peak in the untreated and DMSO groups.

I did not see an explicit data availability statement. Where and when will the data be made available to the community?

*Reviewer #3 (Recommendations for the authors):*

The main concern is the quality of the single cell dataset. In particular, how well the method developed by the authors handles cellular debris that is generated in virtually all dissociation protocols. In fact, the authors explain in their methods that they "centrifuged briefly for 5 sec at 1,000 rcf to separate remaining large debris and single-cells" (line 507) but I doubt that this suffices to remove all particles of debris. The authors could show this by presenting cell cytometry plots of the suspension before and after this brief centrifugation to evaluate what fraction (if any) of the cellular debris they are able to eliminate. Ultimately, the authors should present evidence of the quality of their cell suspensions by showing FACS plots before the 10X encapsulation, and, if there is still considerable cell debris left, use FACS to purify whole cells. Furthermore, in a live cell dissociation cells begin to die quickly. These can be removed by FACS similarly, but the authors do no this. Altogether, this puts into question the author's whole experimental pipeline. The predicted effects of a low quality cell dissociation would be to find genes expressed everywhere, which is precisely what the authors claim for ES products.

Futhermore, the authors should strengthen their analysis to show what parts of it come from real biological signal, and what parts can be explained by the dissociation process. The authors later use SoupX to quantify and remove their ambient RNA content but offer little detail on the process. The major argument to claim for specificity of their data is Figure 3G, where they offer a few markers that are indeed specific of cluster 15. However, this does not suffice to convey the point. Of course, there are some markers that are specific, this is already a given since in fact the clustering algorithm has already singled out these cells as a cluster, based on specific expression. The question is not if there is at least some specific signal, but how good is this signal. The process is not black or white, there are infinite levels of grey in between, where single cells get encapsulated with varying levels of cell debris and/or ambient RNA that provide a general background of general gene expression. It is important to evaluate how good is the cell dissociation, and how much this general background is present.

Consistent with the idea of a low quality dataset, a number of unannotated clusters are presented. Importantly this includes cluster 1, the one with more cells. The authors should investigate the identity of these clusters. Do they have specific markers? The authors should offer lists of markers for all their clusters, including the unannotated ones. The readers would like to see a gene annotation (name of protein / best blast hit) as well as a feature plot. The most likely explanation of these central cluster 1 unannotated cells is that these arise from cellular debris/ambient RNA. The other clusters (emerging from this central cluster) are then those cell barcodes that received debris/ambient RNA together with a cell. This can be evaluated by looking at the markers, to find out if there are only general markers in the central cluster or a cell type identity emerges.

Lastly, the dataset has very low UMI and gene counts per cell, again indicating a relative lack of quality of the dataset. The authors should discuss this in the context of other nematode cell atlases.

---

## [Author Response]

Essential revisions:1) The authors use single-cell transcriptomics without FACS, which is problematic. The main issue is that single-cell transcriptomic methods are so powerful that even without FACS meaningful biological cell type identities emerge. However, the authors go further to claim that secreted products come from a variety of cell types, and this is something that could come from the lack of FACS and a general background signal. The authors will need to validate the quality of their dataset despite the lack of FACS or provide alternative sources of information, be it a new dataset or other validation experiments. Alternatively, they need to tone down the claims about broad unspecific expression and critically discuss the caveats above. The authors will need to make a better effort in explaining the quality (or lack thereof) of their dataset, for the readers to be able to interpret the data.

We have outlined reasons to support our confidence that the data presented meets or exceeds the standards of quality and bioinformatics rigor with respect to the established model nematode literature. We disagree that single-cell transcriptomics without FACS is problematic and that caveats associated with this approach, one that is well-established in the *C. elegans* literature, are not largely addressed through downstream quality control and cautious bioinformatics pipelines. We point to (1) the efforts of our group and others highlighting current difficulties and caveats associated with nematode cell FAC pre-sorting, (2) how our computational pipeline is conservative in its filtering of background signal and poor quality cells, (3) how bulk RNA-seq of a cell population mapped to a defined cell cluster, and (4) how the differential distribution of transcripts associated with secreted products is not at all consistent with a global background signal. We elaborate on several of these points below and provide additional details and analysis in other responses.

We originally intended to utilize FACS to enrich for viable cells prior to sequencing. However, we observed dramatically low levels of cell viability in the sorted cell populations. This phenomenon has also been observed and reported with *C. elegans* single-cell preps (Preston et al. 2019) resulting in the omission of FACS prior to 10x encapsulation for the majority of available *C. elegans* single-cell transcriptomic atlases. Cuticular and other debris in these *C. elegans* single-cell dissociations was not problematic in downstream analyses. A recent *C. elegans* scRNA-seq study reports successfully using FACS prior to 10x encapsulation to enrich for somatic adult cells but were only able to complete the experiments by methanol-fixing and rehydrating the cells prior to FACS and encapsulation, a method not recommended by 10x Genomics at the time our study was completed. It is also just as reasonable to believe that stressing nematode cells through additional processing steps in the way of sorting and fixation-rehydration can introduce artifactual transcriptional states.Our tissue dissociation protocol and scRNA-seq computational analysis pipeline are comparable to published *C. elegans* methods and data. Modifications to these established pipelines only increase the stringency of our quality control and caution in our interpretation. We consider cells with a total UMI count between 100-1,800 UMIs despite several *C. elegans* studies having shown that several neurons are omitted by the standard 100 UMI lower threshold, to further limit undesired background signal. An upper threshold of 1,800 UMIs was chosen as a method to thwart doublet inclusion since the doublet rate when loading the 10X chromium chip could not be accurately estimated. Our estimates of background signal and correction using SoupX were comparable to *C. elegans* (4.15-13.56% with a mean of 8.01%) at 10-13%. Lastly, several *C. elegans* datasets omit cells with >20% mitochondrial representation. We chose a conservative approach and omitted cells that had >10% transcripts associated with mitochondrial genes. Together, these approaches give us confidence that the signal we are analyzing is true biological signal as opposed to background noise that arose from a poor tissue dissociation protocol.We utilized flow cytometry to sort *B. malayi* microfilariae cell populations directly into TRIzol for bulk RNAseq. These populations were defined by cell size and granularity in order to capture the characteristically large secretory cell in microfilariae. The gene transcripts from this pooled cell population mapped primarily to a single cluster in our scRNA-seq atlas indicating specificity of our data.We are able to clearly and specifically detect marker genes and transcription factors for the subset of known microfilariae cell types in our dataset with minimal or no detectable spillover into other cell types. The majority of unannotated clusters in our data also have distinguished but unannotated marker genes in the *Brugia malayi* genome therefore limiting our annotation capabilities. In direct reference to our conclusions about the broad and cell type specific expression patterns of secreted antigens, our analysis in *B. malayi* is mirrored in *C. elegans* orthologs providing additional support that our conclusions are based on biological signal and not ubiquitous background signal.

2) If the authors are able to address the concerns about the quality of the dataset or generate new (e.g., FACS sorted) data, they should elaborate on the interpretation of their single cell data explore the new resource in more detail. For example, a more detailed bioinformatics analysis of the secretory products would provide added value to the paper. However, any further interpretations made from their raw data, without assessing or readdressing the quality of the dataset, are not recommended due to concerns regarding the quality of the data.

We believe that we have satisfactorily addressed these concerns and have included more detailed bioinformatics analyses throughout the manuscript, described in the responses below.

Reviewer #1 (Recommendations for the authors):"neuromuscular and other cell types" – there are no neuromuscular cell types (only neuromuscular system), would be better to write neuronal and muscular cell types

We have made this change in the manuscript.

The authors "mined the literature for a comprehensive list of major antigens, vaccine targets, diagnostic markers, and other notable secreted proteins"It would be useful to provide this list with the references as a source data file

Additional source data file (Figure 3-source data 4) has been added and includes both the list of mined targets and their associated references.

The authors give an EC50 value of 51 μM for IVM treatment. Given the relatively low number of concentrations tested and replicates, this seems too precise. Either they should also provide the error of the estimate, or only say that at 50 μM approximately half of the cells died.

The standard error has been added to the text (EC50 = 51 ± 5.16 µM).

Figure 1—figure supplement 2 – check figure, split across two pages, text should go into figure legend

Figure dimensions are now corrected and text has been moved to the figure legend.

What is the difference between Figure 4A and Figure 4 —figure supplement 1? "Transcript abundance of anthelmintic targets across cell clusters" it seems to be the same data plotted differently. This should be clearly explained (proportion vs total reads).

Figure 4A (now Figure 5A in the revised version) shows the average expression of the indicated gene across all clusters and the size of the dot indicates what percentage of the cells within each cluster express the gene. We wanted to provide the readers with the distribution of the gene transcripts across cell clusters. Additional text has been added to the supplemental figure legend to clarify this.

In general, the figure legends for the figure supplements and source data are minimal. These should be expanded to provide more explanation.

Additional clarification and explanatory details have been added to the supplemental figure legends.

Reviewer #2 (Recommendations for the authors):Given the lack of cell lines for detailed functional studies, the ability to culture dissociated cells is exciting. The authors studied the viability in culture of the dissociated cells. Do any of the dissociated cells divide? This can be an important prerequisite for some transformation protocols. Have attempts to transfected them been made? They have been cultured on surface treated with a lectin and standard culture surfaces. Were differences seen between those? Are the secretory cells present in the culture and maintain viability to 42 h? What happens beyond 42 h?

Thank you for these comments. We have made revisions to Figure 6 (Figure 7 in the revised version) to address your questions about cell viability and proliferation. We quantified the viability of cells in culture out to 96 hrs and although we observed a reduction in viability, a significant subset of cells remain viable at this timepoint. We have attempted transfecting these cells using electroporation and several lipofection approaches without immediate success but this is ongoing work. Unfortunately, these cells do not appear to be in a proliferative state based on an EdU proliferation assay (Figure 7—figure supplement 1), which helps explain the difficulty of successful transfection and which will necessitate optimization of other approaches. Secretory cells are present in single-cell suspensions and were captured in our imaging flow cytometry data at the earliest time points, however they are rare (1 cell/mf) and difficult to identify in culture by eye. Additional details regarding the cell culture viability and proliferation results have been added to the revised Results section.

The fact that immunogenic proteins have a broad cell distribution in the mf is interesting and as noted, raises important questions about how these proteins are exposed to the host. A greater discussion of the immunogenic proteins is warranted. For example, could some immunogenic proteins be exposed to the host by dead or dying worms? Is there an argument to be made for focusing more exclusively on those proteins that are actively transcribed by the secretory cell. Perhaps not, but expanding on this topic would be helpful.

Thank you for your thought-provoking comment. It is likely that some immunogenic proteins are exposed to the host by dead or dying worms, while other proteins encounter the host through secretory/excretory tissues and pathways, as cargo in exosome-like vesicles, or other poorly defined mechanisms. The secretory cell is the most obvious interface for regulated secretion in this particular life stage and our data shows that many immunogenic proteins are in fact expressed in this annotated cell type. In light of underdevelopment of other primary organs, we agree that maybe this is an argument for focusing on actively transcribed secretory-cell associated proteins. We have added a sentence in the discussion to this effect.

Despite the impressive effort in generating the data, they are mostly descriptive. Given the novelty and potential value of these data and methods to the community, this is ok, but clearer discussion on the potential for functional studies is needed. In its current form, the functional potential is too vague and in places reads more like an end, rather than a beginning. For example, one of the more exciting findings is the ability to capture the single secretory cell based on size and physical properties. A potential application could be to sequence transcripts this cell to a greater depth or perform proteomics on these cells to provide a more comprehensive repertoire of its secreted products. Or to follow it developmentally? Or to culture it?

We were able to successfully image secretory cells by imaging flow cytometry and capture the transcriptional profile using a FACS-RNA-seq approach based on cell size but there are still several barriers to isolating secretory cells in culture for further investigation. Since there is a single secretory cell per mf, enriching these cells in culture or completing transcriptomic and proteomic studies is still a challenge without the use of a transgenic marker or antibody labeling approach to isolate this cell in culture. We certainly have interests in the proteomic profiling and developmental trajectory of the secretory cell and these are active areas of investigation for our lab and hopefully the basis of future findings.

This representation is used in several figures and it is sometimes unclear what is important. Consider adding boxes or some other way to highlight important trends in the dot plots. Also, for each dot plot, please make sure there are column label abbreviation keys in the figure legend.

Abbreviation keys have been added to the main and supplemental figure legends for figures that include dot plots. Text within figure legend has been added to direct attention to expression patterns of interest.

In panel 1A, should there be mf in the illustration of the filtration column? The legend for panel 1B indicates that this is the pre and post PD10 filtration. In this case, it is clear there are fewer particles. What are these particles? There is no scale on this. Presumably there should be mf present and they would remain at the same concentration, but they are not visible? Panel 1C needs scale indicated as well even though the mf are clear.

Thank you for your comments. The filtration illustration in panel 1A has been modified to more clearly demonstrate the purpose of filtering the mf prior to beginning the dissociation protocol, which is to separate viable microfilariae from dead microfilariae, embryos, and jird cells and tissue in the lavage (noted in the figure legend). The pre- and post PD10 filtration images in panel 1B are binary image representations of filtrate containing cells/embryos/tissue debris strained with a CellTox viability dye. Viable microfilariae do not stain with CellTox and are therefore not observed in the images. This is a well of a 96 well plate and that is now reflected in the figure legend to provide a sense of scale. Panel 1C has been updated to include a scale bar.

Panel 2B is largely redundant with 2A. As stated above, I think text and number labels with circles around clusters would improve the clarity of 2A and make 2B unnecessary.

Thank you for your feedback. Figure 2 has been updated to improve the clarity of 2A, with cell cluster annotations directly plotted, and to remove the redundancy of 2B. Individual transcript distribution maps for these markers are represented in Figure 2 Figure Supplement 2 (now Figure 2—figure supplement 3).

Legend for Figure 3F should include a legend for the column labels

An abbreviation key has been added to the figure legend.

Panel 3G could be streamlined. I would consider a more compact simpler representation of the data, in line with how it is discussed in the paper. The larger format as it stands now could be moved into the supplement. The number or specific members of the C2H2 Zn finger family members is not discussed in the text.

Thank you for this recommendation and Figure 3 has been modified to omit panel 3G. Additional computational analysis specifically focusing on the secretory cell has been added as an additional figure (Figure 4), which now includes a complete analysis of C2H2 Zn finger transcription factors based on *C. elegans* orthologs. These transcription factors are enriched in the secretory cell so we felt it was important to include this analysis among the main figures as well as additional discussion within the text.

In Panel 5A, are only cells >10^5 in the violet channel considered viable? Either indicate in the text and/or on the graph what the cutoff is since the methanol fixed cells have signal just below 10^5 overlapping where there is a second lower peak in the untreated and DMSO groups.

Thank you for catching this detail. Gated cells with a Calcein Violet-AM intensity greater than 10^5^ were considered viable, and this information has been added to the methods sections where applicable.

I did not see an explicit data availability statement. Where and when will the data be made available to the community?

A Data and Code Availability section is present in the Methods section which included a GitHub repository and NIH BioProject records to allow people to access the raw data and code used to process the data. Unfortunately the size of the processed data was too large to deposit into the GitHub repository so we have deposited the processed data objects (Seurat and AnnData) into a Zenodo repository for easy download by the community. We understand not everybody has the ability to explore this data in R or by Python packages so we have created a RStudio Shiny application to easily explore the data in various capacities. All of these improvements to data availability are reflected in the updated Data and Code Availability subsection of the Methods section.

Reviewer #3 (Recommendations for the authors):The main concern is the quality of the single cell dataset. In particular, how well the method developed by the authors handles cellular debris that is generated in virtually all dissociation protocols. In fact, the authors explain in their methods that they "centrifuged briefly for 5 sec at 1,000 rcf to separate remaining large debris and single-cells" (line 507) but I doubt that this suffices to remove all particles of debris. The authors could show this by presenting cell cytometry plots of the suspension before and after this brief centrifugation to evaluate what fraction (if any) of the cellular debris they are able to eliminate. Ultimately, the authors should present evidence of the quality of their cell suspensions by showing FACS plots before the 10X encapsulation, and, if there is still considerable cell debris left, use FACS to purify whole cells. Furthermore, in a live cell dissociation cells begin to die quickly. These can be removed by FACS similarly, but the authors do no this. Altogether, this puts into question the author's whole experimental pipeline. The predicted effects of a low quality cell dissociation would be to find genes expressed everywhere, which is precisely what the authors claim for ES products.Futhermore, the authors should strengthen their analysis to show what parts of it come from real biological signal, and what parts can be explained by the dissociation process. The authors later use SoupX to quantify and remove their ambient RNA content but offer little detail on the process. The major argument to claim for specificity of their data is Figure 3G, where they offer a few markers that are indeed specific of cluster 15. However, this does not suffice to convey the point. Of course, there are some markers that are specific, this is already a given since in fact the clustering algorithm has already singled out these cells as a cluster, based on specific expression. The question is not if there is at least some specific signal, but how good is this signal. The process is not black or white, there are infinite levels of grey in between, where single cells get encapsulated with varying levels of cell debris and/or ambient RNA that provide a general background of general gene expression. It is important to evaluate how good is the cell dissociation, and how much this general background is present.Consistent with the idea of a low quality dataset, a number of unannotated clusters are presented. Importantly this includes cluster 1, the one with more cells. The authors should investigate the identity of these clusters. Do they have specific markers? The authors should offer lists of markers for all their clusters, including the unannotated ones. The readers would like to see a gene annotation (name of protein / best blast hit) as well as a feature plot. The most likely explanation of these central cluster 1 unannotated cells is that these arise from cellular debris/ambient RNA. The other clusters (emerging from this central cluster) are then those cell barcodes that received debris/ambient RNA together with a cell. This can be evaluated by looking at the markers, to find out if there are only general markers in the central cluster or a cell type identity emerges.

We understand Reviewer #3’s concern over the quality of this single-cell dataset and we certainly share their values in publishing quality and accurate data. We believe we’ve made a very strong case through multiple lines of evidence that the data presented in this paper meets or exceeds field standards for model nematode (*C. elegans*) datasets despite the added complication of working with a parasitic species. We have provided more detailed answers to more specific critiques below:

FACS pre-sorting: We partly addressed the comment about FACS sorting prior to 10x encapsulation earlier in this response. Reviewer #3 is correct that brief centrifugation does not remove all background debris but it does remove large debris like sheath husks and partially digested worm bodies that threaten to clog downstream instrumentation (i.e. flow cytometer, 10X Genomics microfluidic device, etc.). We trialled running cell suspensions through FACS prior to 10X encapsulation to enrich for viable cells, but realized that cells do not survive the FACS process despite using the most gentle instrument settings (see Figure_1—figure supplement 2). This obstacle has also been encountered in the published *C. elegans* single-cell literature, where it is noted that cells struggle to survive the sorting process and viability quickly diminishes in the few cells that do survive the process (Preston et al. 2019). Single-cell transcriptomes from *C. elegans* studies that omit FACS prior to 10x encapsulation are high quality and a significant number of gene expression patterns observed in the scRNA-seq data have been validated in whole-tissue *C. elegans* experiments. We would also argue that omitting FACS prior to encapsulation allowed us to capture the transcriptional state of *B. malayi* mf cells nearer to the time of tissue dissociation. We proceeded without FACS with the understanding that stringent computation filtering and analysis would be required in order to “cut through the noise”, following well-established standards in the field.

Biological signal: The major nematode cell types are represented and differentiated in our global dataset and enriched for expected genes orthologous to established tissue-specific markers in *C. elegans*. We are confident in our specific annotation of these more obvious cell types in the dataset. The annotation of the secretory cell, which we would not expect to be associated with pan-nematodal markers, required additional experiments. We therefore believe our annotation approach is conservative. We have provided additional details about the use of SoupX to filter ambient transcripts and background noise. Some of the critiques the reviewer has made are generically true of all single-cell datasets in non-model organisms, and these caveats are acknowledged. With more sequencing runs, we will be able to update and refine annotations as needed, which is the standard approach across organisms and tissues. This dataset represents an important starting point towards a comprehensive map.

We have further evaluated our dataset using orthogonal approaches. First, we applied a pseudobulk method for cell pattern clustering. The primary annotated cell types in our dataset clustered the same way using this alternative algorithm. Second, we analyzed the pattern of transcription factor expression across our cell clusters and we see global and cluster-specific TF expression patterns that correspond to known expression patterns in other organisms. These datasets and their associated figures are now included in the revised manuscript.

We would like to address the reviewer’s comments about the ‘surprising’ distribution of E/S products across cell types. This does necessarily surprise us. E/S products as sampled by proteomic studies of parasite culture media include mostly proteins without signal peptides and are understood to include proteins from the parasite surface and various orifices. We have mapped a similar thing using replicated spatial transcriptomics approaches in adult stage parasites (Airs et al., PLoS Pathogens 2022). In fact, *C. elegans* orthologs of prominent filarial antigens actually exhibit similar patterns of expression. For example, *far-1* and *unc-15* are sequestered to muscle cell types, *pghm-1* is expressed in neuronal cells, and *mif-1*, *tct-1*, and *tpi-1* are widely expressed across all cell types in both *C. elegans* and *B. malayi*. These mirrored observations in expression patterns across *C. elegans* and *B. malayi* strongly support our capture of real biological signals.

Unannotated cells: Several clusters remain unannotated in our data. Some display cluster-specific gene expression but we do not have enough information about the microfilariae cell types (or the unique biology of filariae) to confidently assign specific identities. Others likely reflect the biological reality of the microfilariae stage. Microfilariae are a post-embryonic and pre-larval development state containing a mixture of terminal and undifferentiated cell types. Microfilariae lack several prominent structures including a complete alimentary canal. These pre-larvae circulate in a suspended developmental state until they are ingested by a mosquito vector and transition to an invertebrate environment, initiating rapid growth and molting events. We therefore expected a larger mass of likely undifferentiable and potentially dormant cell types (e.g., Cluster 1) in the global map of gene expression. These clusters also have low gene and UMI counts per cell, which is in keeping with undifferentiated cells in some other species (e.g., planarian and schistosome neoblasts). For these reasons, we only assigned annotations for major cell types where clearly supported, understanding that cell-to-cell comparisons of clade III and clade V nematodes are more risky for deeper annotation. A Supplementary file containing differentially expressed genes for each cluster has been added to the manuscript, which could help with future annotations of these more ambiguous clusters.

Lastly, the dataset has very low UMI and gene counts per cell, again indicating a relative lack of quality of the dataset. The authors should discuss this in the context of other nematode cell atlases

Nematode cells are comparatively much smaller than mammalian cells and will therefore have lower UMI and gene counts compared to the majority of published scRNAseq studies in the literature. In the context of other nematodes, our data is within the range of expression seen in *C. elegans* datasets when we consider the developmental stages of the nematodes (see table below). If we omit the cells in cluster 1 with the assumption that these are undifferentiated cells and therefore low expression, our data has a mean of ~411 UMI/cell and ~319 genes/cell. Additionally, a sizeable portion of the differentiated cells at this development state are neuronal, which are reported to be very low expressing cells in *C. elegans* and are often accidentally omitted from analysis when using standard scRNA-seq UMI/gene expression cutoffs. Lastly, UMI and gene counts per cell are partially dependent on the depth of sequencing, which differs across studies. Our libraries were sequenced to 60-70% saturation and while it’s possible that greater depth would marginally improve these statistics, our dataset aligns with published studies and is adequate to differentiate major cell types.

**Author response table 1. sa2table1:** 

Paper	Stage	UMIs/cell	genes/cell
Preston et al., 2019	embryo	156	52
This study	mf	267 (median)	230 (median)
Cao et al., 2017	L2	575 (median)1,121 (mean)	431 (mean)
Taylor et al., 2021	L4	893 (median)	321 (median)
Roux et al., 2022	Adult	2,175	644